

# Global change effects on decomposition processes in tidal wetlands: implications from a global survey using standardized litter

## Authors

Peter Mueller*[1], Lisa M. Schile-Beers[2], Thomas J. Mozdzer[3], Gail L. Chmura[4], Thomas Dinter[5], Yakov Kuzyakov[5,6], Alma V. de Groot[7], Peter Esselink[8,9], Christian Smit[9], Andrea D'Alpaos[10], Carles Ibáñez[11], Magdalena Lazarus[12], Urs Neumeier[13], Beverly J. Johnson[14], Andrew H. Baldwin[15], Stephanie A. Yarwood[15], Diana I. Montemayor[16], Zaichao Yang[17], Jihua Wu[17], Kai Jensen[1], and Stefanie Nolte[1]

## Affiliations

1) Applied Plant Ecology, Biocenter Klein Flottbek, Universität Hamburg, Ohnhorststraße 18, 22609 Hamburg, Germany
2) Smithsonian Environmental Research Center, 647 Contees Wharf Rd, Edgewater, MD, 21037, USA
3) Bryn Mawr College, Department of Biology, 101 N. Merion Ave, Bryn Mawr, PA, 19010, USA
4) Department of Geography, McGill University, 805 Sherbrooke St W, QC H3A 0B9, Canada
5) Dept. of Soil Science of Temperate Ecosystems, University of Goettingen, Büsgenweg 2, 37077 Göttingen, Germany
6) Institute of Environmental Sciences, Kazan Federal University, 420049 Kazan, Russia
7) Wageningen Marine Research, Wageningen University & Research, Den Helder, Ankerpark 27, 1781AG, The Netherlands
8) PUCCIMAR, Boermarke 35, 9481 HD, Vries, The Netherlands
9) Conservation Ecology Group, Groningen Institute for Evolutionary Life Sciences, University of Groningen, P.O. Box 11103, 9700 CC, Groningen, The Netherlands
10) Department of Geosciences, University of Padova, Via Gradenigo 6, Padua 35131 Italy
11) IRTA Aquatic Ecosystems, Carretera Poblenou Km 5.5, 43540 Sant Carles de Ràpita, Catalonia, Spain
12) Department of Plant Taxonomy and Nature Conservation, University of Gdansk, ul. Wita Stwosza 59, 80-308 Gdansk, Poland
13) Institut des sciences de la mer de Rimouski, Université du Québec à Rimouski, 310 allée des Ursulines, Rimouski QC G5L 3A1, Canada
14) Department of Geology, Bates College, 214 Carnegie Sciences Building, Lewiston, ME, 04240, USA
15) Department of Environmental Science & Technology, University of Maryland, College Park, MD 20742
16) Instituto de Investigaciones Marinas y Costeras (IIMyC), CONICET, UNMDP, Mar del Plata, Argentina
17) Ministry of Education Key Laboratory for Biodiversity Science and Ecological Engineering, Institute of Biodiversity Science, Fudan University, Shanghai 200438, PR China

*Corresponding author:
Dr Peter Mueller; peter.mueller@uni-hamburg.de



## Abstract

Tidal wetlands, such as tidal marshes and mangroves, are hotspots for carbon sequestration. The preservation of organic matter (OM) is a critical process by which tidal wetlands exert influence over the global carbon cycle and at the same time gain elevation to keep pace with sea-level rise

(SLR). The present study provides the first global-scale field-based experimental evidence of temperature and relative sea level effects on the decomposition rate and stabilization of OM in tidal wetlands. The study was conducted in 26 marsh and mangrove sites across four continents, utilizing commercially available standardized OM. While effects on decomposition rate per se were minor, we show unanticipated and combined negative effects of temperature and relative sea level on OM

stabilization. Across study sites, OM stabilization was 29% lower in low, more frequently flooded vs. high, less frequently flooded zones. OM stabilization declined by ~90% over the studied temperature gradient from 10.9 to 28.5°C, corresponding to a decline of ~5% over a 1°C-temperature increase. Additionally, data from the long-term ecological research site in Massachusetts, US show a pronounced reduction in OM stabilization by >70% in response to

simulated coastal eutrophication, confirming the high sensitivity of OM stabilization to global change. We therefore provide evidence that rising temperature, accelerated SLR, and coastal eutrophication may decrease the future capacity of tidal wetlands to sequester carbon by affecting the initial transformations of recent OM inputs to soil organic matter.





## 1 Introduction

Tidal wetlands, such as marshes and mangroves, provide a wide array of ecosystem services that have been valued at approximately US$ 10,000 per hectare and year, making them some of the most economically valuable ecosystems on Earth (Barbier et al., 2011; Kirwan and Megonigal, 2013).

Yet, tidal wetlands are threatened and vulnerable ecosystems, experiencing pronounced loss though global-change impacts, such as land use (Pendleton et al., 2012), and accelerated sea-level rise (SLR; Craft et al., 2009; Crosby et al., 2016). In recent years, carbon sequestration has increasingly been recognized as an ecosystem service of tidal wetlands (e.g. Chmura et al., 2003; Mcleod et al., 2011). Tidal wetlands are efficient long-term carbon sinks, preserving organic matter (OM) for

centuries to millennia. Here, high rates of OM input (from both autochthonous and allochthonous production) co-occur with reducing soil conditions and thus slow rates of decomposition, leading to organic carbon sequestration rates that exceed those of most other ecosystem types by orders of magnitude (Mcleod et al., 2011). At the same time, suppressed decomposition and the preservation of OM is a primary process by which many tidal wetlands gain elevation and keep pace with rising

sea level (Kirwan and Megonigal, 2013). Consequently, global changes that might increase OM decomposition in tidal wetland soils not only affect carbon sequestration, but also decrease ecosystem stability against SLR. It is therefore critical to identify global change factors that affect the transformation of organic inputs to stable soil OM (SOM) in tidal wetlands and to assess the magnitude of their effects.

There are multiple methods for assessing factors that influence carbon sequestration, including direct measurements of plant production, organic carbon stocks, accretion, and decomposition rates. Litter-bag techniques assessing the weight loss of plant material over time are probably the easiest way to measure decomposition rates in situ and have been widely used since the 1960s (Prescott, 2010). Global-scale assessments of litter decomposition have been conducted

as both meta-analyses (e.g. Zhang et al., 2008) and as inter-site studies along latitudinal gradients





(Berg et al., 1993; Torfymow et al., 2002; McTiernan et al., 2003; Cornelissen et al., 2007; Powers

et al., 2009) in order to assess effects of climate parameters, mostly with focus on temperature and

moisture gradients, on decomposition rate. Besides abiotic or climate effects, these studies could

also identify litter quality itself as an important predictor for decomposition rate (Zhang et al.,

100     2008).

Relationships between single climate or litter-quality parameters and decomposition rate

often are not linear. Instead, complex interactions between litter quality and climate parameters

seem to control litter decomposition (Zhang et al., 2008), creating challenges in separating climate

from litter-quality effects and predicting the relevance of potential global-change drivers for

decomposition rate. In order to separately assess environmental or climate effects on litter

decomposition at a global scale, it is therefore necessary to standardize litter quality in inter-site

studies. However, implications of litter-decay data for carbon sequestration need to be considered

cautiously, as the link among litter-decomposition rate, SOM formation, and ultimately carbon

sequestration is not straightforward (Prescott, 2010; Cotrufo et al., 2013): Because plant tissues are

not resistant to decay per se, it is critical to understand their biogeochemical transformation into

stable compounds that leads to the formation of SOM (*i.e.* stabilization) rather than understanding

the pace at which early stages of decomposition proceeds (Prescott, 2010; Castellano et al., 2015;

Haddix et al., 2016).

Keuskamp and others (2013) developed an efficient approach for studying litter

decomposition and OM transformation at a global scale, using commercially available tea as

standardized material. Their Tea Bag Index (TBI) approach is based on the deployment of two types

of tea that considerably differ in their OM quality. The method allows for the determination of the

decomposition rate constant (in the following referred to as *decomposition rate* or *k*), as in classic

litter-bag approaches, and a stabilization factor (in the following referred to as *stabilization* or *S*),

which describes the fraction of labile and decomposable OM that becomes stabilized during



deployment.

In the present study, we assessed the impacts of multiple global-change factors – warming, sea-level rise (SLR), salt-water intrusion, and coastal eutrophication – on both OM decomposition rate and stabilization in tidal wetland soils by conducting a worldwide field study using

standardized litter. First, by covering a large temperature gradient of $\Delta T >15\ °C$ across sites, we aimed to capture temperature effects on OM decomposition and stabilization, thereby improving our process-level understanding on how global warming affects carbon turnover and ultimately sequestration in tidal wetlands. Second, by conducting paired measurements in both high- and low-elevated zones of tidal wetlands worldwide, we were aiming to gain insight into potential effects of

accelerated relative SLR on carbon turnover. Despite the dominant paradigm that decomposition is inversely related to flooding, the existing literature on hydrology and SLR effects on OM decomposition in tidal wetlands yields equivocal results, which is often due to the overriding effect of OM quality on decomposition rate (Hemminga and Buth, 1991; Kirwan et al., 2013; Mueller et al., 2016). Additionally, by expanding our study to include fresh and brackish sites, we anticipated

to capture the effects of salt-water intrusion into brackish and fresh systems, which is likely to affect decomposition processes in tidal wetlands (Morrissey et al., 2014). Specifically, high concentrations of dissolved sulfate in seawater, acting as an alternative terminal electron acceptor, can enhance anaerobic microbial metabolism in systems with lower salinity (Megonigal et al., 2004; Sutton-Grier et al., 2011). Lastly, we used the long-term ecological research site of the TIDE project in

Massachusetts, US (Deegan et al., 2012) to experimentally assess both the effects of coastal eutrophication and – with respect to SLR-driven increases in flooding frequency – the relevance of nutrient delivery through floodwater for the early stages of OM decomposition in tidal wetlands.

## 2 Methods

### 2.1 Study sites and experimental design



The study was conducted in 26 tidal wetlands during the 2015 growing season (Figure 1, Table 1).

Nine sites were situated along the European coasts of the North Sea, Mediterranean, and Baltic, ten

sites were located along the East and West coasts of North America including the St. Lawrence

Estuary, Bay of Fundy, Chesapeake Bay, and San Francisco Bay, and four mangrove sites were

situated along the Caribbean coast of Central America in Belize and Panama. Additionally, one

Chinese site (Yangtze Estuary) and one Argentinian site were included in our study. Sixteen of the

sites were salt marshes, six were tidal freshwater and brackish sites, and four sites were mangroves.

At 22 sites, we compared high and low elevated zones, which were characterized by distinct plant

species compositions (i.e. different communities in high vs. mid vs. low marshes) or by different

stature of mangroves (i.e. dwarf vs. fringe phenotypes). We used relative elevation as a site-specific

proxy for relative sea level. By doing so, we did not capture the actual variability in the tidal

inundation regime across our study sites as these vary in absolute elevation and in elevation relative

to mean high water. Finally, we included the long-term experimental site of the TIDE project in

Massachusetts, US to assess effects of nutrient enrichment on litter-decomposition rate and

stabilization. Through nitrate additions to the incoming tides on at least 120 days per year, nutrient

enriched areas at the TIDE project site receive floodwater with 10-15 fold increased nitrogen (N)

concentrations compared to reference areas since 2004. From 2004-2010 also phosphate was added

to the floodwater, however, this has been discontinued because creek water P concentrations are

high enough to prevent secondary P limitation through N enrichment (details in Deegan et al., 2012;

Johnson et al., 2016).

Decomposition rate and stabilization were measured by deploying tea bags in ten points per

zone or treatment within a site (n=10). Spacing between replicates within a zone or treatment was

≥2 m. However, as sites differed considerably in their areal extent, the distribution and thus spacing

between points had to be adjusted to be representative for the given system. Temperature for the

period of deployment was measured at site or temperature data was obtained from the online service



of *Accuweather* (accuweather.com; accessed 12/25/2016) for locations within a distance of 15 km

        to the site for most sites, but not further than 60 km for some remote sites.

        *2.2      Decomposition rate and stabilization measurements*

        Decomposition rate (*k*) and stabilization factor (*S*) were assessed following the *Tea Bag Index*

protocol (Keuskamp *et al.*, 2013). Briefly, at each point two nylon tea bags (200 µm mesh size), one

        containing green tea (EAN: 8 722700 055525; Lipton, Unilever + PepsiCo, UK) and one containing

        rooibos (8 722700 188438, Lipton, Unilever + PepsiCo, UK), were deployed as pairs in ~8 cm soil

        depth, separated by ~5 cm. The initial weight of the contents was determined by subtracting the

        mean weight of 10 empty bags (bag + string + label) from the weight of the intact tea bag prior to

deployment (content + bag + string + label). The tea bags were retrieved after an incubation time of

        ~90 days. Upon retrieval, tea bags were opened, tea materials were carefully separated from clay

        particles and fine roots, dried for 48 h at 70°C, and weighed.

                Calculations for *k* and *S* followed Keuskamp et al. (2013):

Eq 1)                              $W_r(t) = a_r\,e^{-kt} + (1-a_r)$

        Eq 2)                              $S = 1 - a_g / H_g$

        Eq 3)                              $a_r = H_r\,(1-S)$


        $W_r(t)$ describes the substrate weight of rooibos after incubation time (*t* in days), $a_r$ the labile and $1-a_r$

        the recalcitrant fraction of the substrate, and *k* is the decomposition rate constant. *S* describes the

        stabilization factor, $a_g$ the decomposable fraction of green tea (based on the mass loss during

        incubation) and $H_g$ the hydrolysable fraction of green tea. The decomposable fraction of rooibos tea





is calculated in Eq 3 based on its hydrolysable fraction ($H_r$) and the stabilization factor $S$. With

$W_r(t)$ and $a_r$ known, $k$ is calculated using Eq 1.

In accordance with Keuskamp et al. (2013), extractions for determination of the

hydrolysable fractions of green and rooibos tea followed Ryan et al. (1990). However, instead of

using Ryan's *forest products protocol* we conducted the alternative *forage fiber protocol* for the

determination of the hydrolysable fraction. Briefly, 1 g of dried tea material (70°C for 24 h) was

boiled in cetyltrimethyl ammonium bromide (CTAB) solution (1 g CTAB in 100 ml 0.5 M $H_2SO_4$)

for 1 h (Ryan et al., 1990; Brinkmann et al., 2002). The extract was filtered through a 16-40-µm

sinter filter crucible (Duran, Wertheim, Germany) using a water-jet vacuum pump and washed with

150 ml of hot water followed by addition of acetone until no further de-coloration occurred

(Brinkmann et al., 2002). The remaining material was left in the sinter, dried for 12 h at 70°C,

cooled in a desiccator and weighed. 20 mL of 72% $H_2SO_4$ was added to the sinter and filtered off

after an incubation of 3 h, followed by washing with hot water to remove remaining acid. The sinter

was dried at 70°C for 12 h, cooled in a desiccator, and weighed to determine the non-hydrolysable

fraction. Finally, the sinter containing the remaining sample was ignited at 450°C for 3 h in order to

determine the ash content of the material.

In addition to the determination of the hydrolysable fraction, we measured total C and N

contents of the tea material using an elemental analyzer (HEKAtech, Wegberg, Germany). The

hydrolysable fraction of both green and rooibos tea was higher than reported in Keuskamp et al.

(2013) (Table 2). However, the determined C and N contents of the tea materials are in agreement

with those reported in Keuskamp et al. (2013) (Table 2). Therefore, deviations from the

hydrolysable fraction as reported previously are likely due to the less conservative extraction

assessment in the present study and not due to actual changes in the quality of the materials.

*2.3    Data Analyses*



For all across-site analyses, mean values of each *site* by *elevation zone* (or *site* by *salinity class)*

combination were used (N=51). Relationships between single parameters and litter decomposition

are often not linear. Instead, critical thresholds seem to exist at which a certain predictor (e.g. mean

annual temperature) becomes influential (Rothwell et al., 2008; Prescott, 2010).

     In the first step of our data analysis, we therefore used classification and regression tree

analysis (CRTA) to identify important predictors for $k$ and $S$. CRTA is a non-parametric procedure

for the step-wise splitting of the data set with any number of continuous or categorical and

correlated or uncorrelated predictor variables (Breiman et al., 1984; Rothwell et al., 2008), and it

has been recommended to identify thresholds and to handle large-scale decomposition data sets

(Rothwell *et al.*, 2008; Prescott, 2010). We conducted CRTA separately for $k$ and $S$ using

temperature, salinity class, tidal amplitude, ecosystem type, soil type, and relative elevation as

predictor variables (Table 1). V-fold cross validation was set at 5 (as commonly conducted,

compare Rothwell et al. (2008)), and the minimum number for observations per child node was set

at n = 4, corresponding to at least two sites or 8% of the total data set.

     To test for correlations between the variables salinity class, temperature, latitude, tidal

amplitude, $k$ and $S$, Spearman rank correlations were used (Table 3). Mann-Whitney U tests were

conducted to test for differences in $k$ and $S$ between marshes and mangroves and between mineral

and organic soil types.

     We tested for linear effects of temperature on $k$ and $S$ across sites, using simple linear

regression analyses (Fig. 2). Two-tailed paired t-tests were used to test for effects of relative

elevation as proxy for relative sea level on $k$ and $S$ (Fig. 3). Subsequent one-tailed paired t-tests

were conducted to test for the same effect within mineral, organic, marsh, and mangrove systems

separately.

     In 21 of our 22 sites where tea bags were deployed in both high and low elevation zones,

replication was sufficient to conduct one-way ANOVA to test for differences in $k$ and $S$ between



zones for each site separately (Fig. S2). We tested for effects of nutrient enrichment on $k$ and $S$ in

the data from the TIDE site (Massachusetts, US) using two-way ANOVA with enrichment

treatment and marsh zone as predictors. All analyses were conducted using STATISTICA 10

(StatSoft Inc., Tulsa, OK, USA).

**3       Results**

*3.1     Temperature effects*

We found no linear (Fig. 2a) or monotonic (Table 3) relationships between temperature and $k$. Also,

CRTA revealed temperature only as a minor predictor for $k$ (Figure S1a). Specifically, temperature

seems to positively affect $k$ in meso-tidal systems only (amplitude >2.1m; Fig. S1a; node 5) with

sites ≥14.5°C during deployment supporting considerably higher rates of decomposition than sites

characterized by lower temperatures. However, this apparent temperature effect was inconsistent

within the group of observations with tidal amplitude >2.1m (Fig. S1a; nodes 13-15). Furthermore,

the majority of sites (65%) are characterized by tidal amplitudes <2.1 m and show no temperature

effect on $k$.

260         In contrast to the temperature response of $k$, $S$ was strongly affected by temperature (Fig.

2b). The significant negative correlation between $S$ and temperature (p < 0.01; $r^2$ = 0.287; Fig. 2b)

agrees well with the CRTA (Fig. S1b). However, CRTA also identified a narrow temperature range

(21.9-23.6°C) in which increasing temperature led to higher stabilization (Fig. S1b; node 11). This

group of observations (n = 5) from the general pattern is also clearly visible in Fig. 2b.


*3.2     Effects of relative sea level*

Paired comparisons of high vs. low elevated zones indicate no consistent effect of relative sea level

on $k$ across sites (p > 0.1; Fig. 3a + c), whereas $S$ was significantly reduced by 29% in low

compared to high elevated zones (p < 0.01; Fig. 3b). This significant reduction of $S$ in low vs. high





elevated zones was consistent across mineral and organic, as well as marsh and mangrove systems

(Fig. 3d). Testing for effects of relative sea level within each site separately revealed that *S* is

significantly reduced by 28-87% in the lower elevated zone in 15 of 21 sites. A significant increase

of *S* in low vs. high elevated zones was found in none of these 21 sites (Fig. S2). In ten of the sites,

we found a significant effect of relative sea level on *k*; with significantly higher *k* in low vs. high

zones in seven sites and significantly lower *k* in low vs. high zones in three sites (Fig. S2).

### 3.3    *Effects of salinity and nutrient enrichment*

We found no significant relationship between salinity class and *k* or *S* (Table 3). Also, CRTA did

not reveal salinity class as an important factor for *k* and *S* (Fig. S1), and no consistent salinity effect

on k and *S* was found when comparing sites of different salinities within single estuarine regions

(Chesapeake, Ebro Delta, Long Marsh, San Francisco Bay; Fig. S3).

The nutrient enrichment treatment at the TIDE project site decreased *S* by 72% in the high

marsh. *S* in the low marsh likewise was similarly low as in the fertilized high marsh and not further

reduced by fertilization (Fig. 4). In contrast, *k* was not responsive to the fertilization treatment in

neither low nor high marsh (Fig. 4).

### 3.4    *Other factors influencing decomposition rate and stabilization*

CRTA revealed tidal amplitude as an important predictor for *k* (Fig. S1a). However, this result

needs to be interpreted cautiously because no linear ($p > 0.68$; $r^2 = 0.004$) or monotonic relationship

(Table 3) existed between tidal amplitude and *k*, and effects of tidal amplitude are not independent

from other factors because strong correlations existed with ecosystem and soil type, temperature,

and latitude (Table 3).

Soil (mineral vs. organic) and ecosystem type (marsh vs. mangrove) did not affect *k* (Table

3, Fig. S1a). In comparison, *S* was lower in mangroves than in marshes and lower in organic than in



mineral systems (Table 3), presumably caused by temperature effects because ecosystem and soil

type did not show up as predictors in CRTA (Fig. S1b).

## 4       Discussion

The findings of the present study cannot demonstrate consistent effects of either temperature or

relative sea level on the decomposition rate of recent OM inputs (commonly assessed as *k* in litter

bag studies) in tidal wetlands. With respect to C sequestration, however, litter-decay data need to be

considered cautiously, as the link among decomposition rate, SOM formation, and ultimately C

sequestration is not straightforward. That is, plant tissues and other fresh OM inputs into an

ecosystem are not resistant to decay per se, and as a consequence, it is critical to understand their

biogeochemical transformation into stable compounds that leads to the formation of SOM (*i.e.*

stabilization) rather than understanding the pace at which decomposition proceeds (Prescott, 2010;

Castellano et al., 2015; Haddix et al., 2016). Here, we also assessed OM stabilization, and in

contrast to decomposition rate, stabilization decreased with temperature and was consistently lower

in low vs. high elevated zones of tidal wetlands. Our study therefore provides indirect evidence that

rising temperature and accelerated SLR could decrease the capacity of tidal wetlands to sequester C

by affecting the initial transformations of recent OM inputs to SOM (i.e. stabilization).

### *4.1     Temperature effects on decomposition processes*

Surprising in the context of basic biokinetic theory, the temperature response of decomposition rate

was weak or not present. Following typical Q10 values for biological systems of 2-3 (Davidson &

Janssens, 2006), *k* should have at least doubled over a gradient of ΔT >15°C. However, findings

from studies conducted at single-marsh to regional scales are not conclusive either, ranging from no

or small (Charles & Dukes, 2009; Kirwan et al., 2014; Janousek et al., 2017) to strong seasonally-

driven temperature effects with a Q10 >3.4 as found within a single site (Kirwan & Blum, 2011).




Although temperature sensitivity of OM types is variable (Craine *et al.*, 2010; Hines *et al.*, 2014;

Wilson *et al.*, 2016), temperature sensitivity of the used TBI materials was sufficiently

demonstrated (Keuskamp et al., 2013). We therefore conclude that other parameters exerted

overriding influence on *k*, mainly masking temperature effects and have not been captured by our

experimental design. For instance, we do not have data on plant-biomass parameters that are

thought to exert strong control on decomposition in tidal wetlands through priming effects (Wolf et

al., 2007; Mueller et al., 2016; Bernal et al., 2017). Likewise, large differences in site elevation and

hydrology could have induced high variability in *k* across sites and masked potential temperature

effects. Indeed, we demonstrate significant but mixed effects of relative sea level on *k* for some

sites; however, we do not have sufficient data on actual site elevation or hydrology to control for

these factors as covariates affecting the temperature effect on *k*.

In contrast to missing or subtle effects of temperature on *k*, OM stabilization was strongly

affected by temperature. Overall, *S* decreased by ~90% over our temperature gradient from 10.9 to

28.5°C, corresponding to a decline of ~5% over a 1°C-temperature increase (Figure 2b). Thus, we

demonstrate a considerable temperature effect on the initial steps of biomass decomposition in tidal

wetlands. This effect, however, is not driven by changes in decomposition rate per se, but – more

importantly – by affecting the transformation of fresh and decomposable organic matter into stable

compounds, with implications for C sequestration (e.g. Cotrufo et al., 2013).

In their global-scale assessment, Chmura et al. (2003) report a negative relationship of soil

organic C density and mean annual temperature within both salt marshes and mangroves. Indeed,

Chmura and colleagues hypothesized stimulated microbial decomposition at higher temperatures to

be the responsible driver for this relationship. Plant production and thus OM input is known to

increase with latitude and temperature in tidal wetlands (Charles & Dukes, 2009; Gedan &

Bertness, 2009; Kirwan *et al.*, 2009; Baldwin *et al.*, 2014), but this increase seems to be more than

compensated by higher microbial decomposition. Working on the same spatial scale as Chmura et



al. (2003), our study strongly supports this hypothesis and provides the mechanistic insight into the

temperature control of OM decomposition as an important driver of C sequestration tidal wetlands.

### 4.2     Relative-sea-level effects on decomposition processes

Flooding and thus progressively lower oxygen availability in soil is supposed to be a strong

suppressor of decomposition (Davidson & Janssens, 2006). Despite this dominant paradigm, we

clearly demonstrate that $k$ is not reduced in low vs. high elevated zones of tidal wetlands (Fig. 3a).

This finding is in accordance with an increasing number of studies demonstrating negligible direct

effects of sea level on decomposition rate in tidal wetland soils (Kirwan et al., 2013; Mueller et al.,

2016; Janousek et al., 2017). A SLR-induced reduction in decomposition rate with positive

feedback on tidal wetland stability seems therefore to be an unlikely scenario. Furthermore, we

show that $S$ is strongly reduced in low vs. high elevation zones, suggesting that the conversion of

recent OM inputs to stable compounds and eventually SOM is in fact lower in more flooded zones

of tidal wetlands. Accelerated SLR consequently seems to yield the potential to decrease SOM

formation and with that C sequestration.

360             This finding and its implication may occur counterintuitive with respect to the often sharp

redox gradients along tidal wetland zonations and with flooding (Davy et al., 2011; Kirwan et al.,

2013; Langley et al., 2013), and the mechanism by which $S$ is decreased in the more flooded zones

is unknown. Because we did not observe consistent salinity effects on $S$ and $k$ in our data (Figs. S1,

S3), we do not suppose that regular exposure of litter to salt water explains the unexpected finding.

Instead, more favorable soil moisture conditions in low vs. high elevated zones could have

decreased OM stabilization if higher flooding frequencies did not induce redox conditions low

enough to suppress microbial activity in the top soil. In support of this, flooding-frequency induced

changes in moisture conditions have been reported as primary driver of surface litter break down,

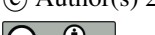



leading to more than four-fold increased litter mass loss in low vs. high marsh zones of a New

Jersey salt marsh (Halupa & Howes, 1995).

Additionally, greater nutrient availability and less nutrient-limited microbial communities in
more frequently flooded zones could have contributed to this effect (Deegan et al., 2012; Kirwan et
al., 2013). Strong effects of both high quality marine-derived OM and nutrient amendments on
microbial structure and activity have been reported (Deegan et al., 2012; Keuskamp et al., 2015a;

Kearns et al., 2016; Mueller et al., 2017), suggesting that regular marine OM and nutrient inputs in
more frequently flooded zones can positively affect decomposition.

### 4.3    *Nutrient enrichment reduces stabilization – insights from the TIDE project*

In addition to our global survey of early-stage decomposition processes in tidal wetlands

worldwide, we included the long-term ecological research site of the TIDE project in
Massachusetts, US to experimentally assess both the effects of coastal eutrophication and the
relevance of nutrient delivery through floodwater for OM decomposition in tidal wetlands.
Important for our argument that decomposition may be favored by higher nutrient availability in
low elevated, more frequently flooded zones, we observed a strong reduction (>70%) of $S$ by

nutrient enrichment in the high marsh, whereas $S$ in the low marsh likewise was low as in the
fertilized high marsh and not further reduced by fertilization (Fig. 4). Johnson et al. (2016)
demonstrate that nutrient enriched high-marsh plots of the TIDE project receive $19\pm2$ g N m$^{-2}$ yr$^{1}$,
approximately 10-times the N load of reference high-marsh plots ($2\pm1$ g N m$^{-2}$ yr$^{-1}$; mean$\pm$SE), thus
explaining the strong treatment effect observed in the high marsh. In accordance with low

stabilization in the reference low marsh, which is equally low as the nutrient enriched high marsh,
reference plots of the low marsh receive $16\pm4$ g N m$^{-2}$ yr$^{1}$, the same high N load as the enriched
high-marsh plots. Surprisingly, however, N loads of $171\pm19$ g N m$^{-2}$ yr$^{1}$ in the enriched low-marsh
plots do not result in additional reduction of $S$ compared to the reference low marsh (Fig. 4). These

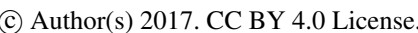

findings suggest that microbial communities of the high marsh are N limited, and that N additions to

a certain level can stimulate early OM decomposition and thus reduce stabilization. The missing

effect of N loads exceeding 16 g m$^{-2}$ yr$^{1}$ on stabilization in the low marsh indicates that microbial

communities are less or not N limited due to the naturally greater nutrient availability. The findings

of the TIDE project therefore support our concept that higher nutrient availability and less nutrient-

limited microbial communities in more frequently flooded zones could have contributed to the

observed reduction of OM stabilization in low vs. high elevated zones of tidal wetlands in our

global assessment.

Although our conclusions on effects of nutrient enrichment on OM decomposition are based

on the findings of a single field experiment only, our study adds to a growing number of reports

illustrating the impact of coastal eutrophication on tidal wetland C cycling (Morris & Bradley,

1999; Deegan et al., 2012; Kirwan & Megonigal, 2013; Keuskamp et al., 2015b). At the same time,

however, we highlight the need to improve our understanding of coastal eutrophication in

interaction with other global changes, particularly accelerated SLR and concomitant changes in

flooding frequency, on the cycling of both labile and refractory C pools in order to predict future

stability of tidal wetlands.


### 4.4   *Methodological considerations*

The quality of OM (i.e. its chemical composition) is a key parameter affecting its decomposition.

As the quality of the TBI materials differ from that of wetland plant litters, we did not expect to

capture precise and absolute values for wetland litter break down in this study. Instead, we used the

Tea-Bag Index to characterize the decomposition environment by obtaining a measure for the

potential to decompose and stabilize the deployed standardized material. Standardized approaches

like this, or also the cotton-strip assay (e.g. Latter and Walton, 1988), are useful to separate the

effects of environmental factors other than OM quality on decomposition processes and to assess



their relative importance. Otherwise, complex interaction effects of the abiotic environment and

OM quality make it difficult to predict the relevance of certain environmental factors for

decomposition processes, potentially masking the effects of important global-change drivers

(reviewed in Prescott, 2010).

Stabilization is thought to be a key parameter for capturing changes in decomposition with

consequence for C sequestration. Indeed, Keuskamp et al. (2013) demonstrate that $S$, as determined

by the TBI, is significantly related with the C sequestration potential of an ecosystem as defined by

FAO (2000). In the present study, however, a large percentage of observations showed relatively

low values for $S$, although tidal wetlands are known to act as particularly effective C sinks (Mcleod

et al., 2011). Based on the $S$ values obtained from initial calculations using the hydrolysable

fractions suggested by Keuskamp et al. (2013), a large number of observations in fact yielded a

negative $S$ (data not shown). $S$ becomes negative when the mass loss from green tea is greater than

the predicated maximum loss based on its hydrolysable fraction. At least two processes could have

caused or contributed to this result: First, we demonstrate that $S$ is indeed reduced in low vs. high

elevated zones across our study sites, indicating that redox conditions in the top soil of tidal

wetlands are at least often not low enough to hamper decomposition processes. As a consequence,

the relatively high top-soil moisture of tidal wetlands provide favorable conditions for

decomposition, following typical moisture-decomposition relationships as demonstrated for

terrestrial ecosystems (e.g. Curiel Yuste et al., 2007), and $S$ should at least not expected to be high

in the top-soil environment of tidal wetlands. Potentially, moisture and nutrient supply are even

high enough to allow for considerable break down of non-hydrolysable compounds within three

months of deployment, such as lignin (Berg & McClaugherty, 2003; Knorr et al., 2005; Feng et al.,

2010; Duboc et al., 2014). Second, different protocols and methods to determine hydrolysable and

non-hydrolysable fractions of plant materials exist and lead to variable results. The hydrolysable

fraction of plant materials can consequently be over- or underestimated depending on method,



protocol, and type of sample material. The use of the slightly higher hydrolysable fractions we

determined for calculations of the TBI parameters effectively eliminated negative $S$ values. In that

regard, using the values obtained from the alternative protocol given in Ryan et al. (1990) seemed

more reasonable in our study. Although direction and size of reported effects on $S$ and $k$ in the

present study are almost independent from the hydrolysable fraction used for calculations, further

research is required to improve our estimates of the hydrolysable fractions in TBI materials.


### 4.5    Implications

While awareness about potential global-warming impacts on OM preservation and their resulting

threat to future tidal wetland stability has been raised (Kirwan & Mudd, 2012), concepts on the

vulnerability of tidal wetlands to accelerated SLR mainly focus on plant-productivity responses and

their biophysical feedbacks (Kirwan et al., 2016). Potentially negative effects of accelerated SLR on

OM preservation were thus far overlooked, probably because stimulation of decomposition

processes through increasing flooding is counterintuitive (Mueller et al., 2016). Here, we provide

evidence that accelerated SLR is unlikely to slow down the decomposition rate of fresh OM inputs

and additionally may strongly decrease OM stabilization and thus potentially the fraction of net

primary production and other OM inputs to stable SOM.

This study addresses the influence of temperature, relative sea level, and coastal

eutrophication on the initial transformation of biomass to SOM, and it does not encompass their

effects on the existing SOM pool. However, aspects of $S$ and $k$ are key components of many tidal

wetland resiliency models (Schile et al., 2014; Swanson et al., 2014) that have highlighted the

critical role of the organic contribution to marsh elevation gain. Thus, combined negative effects of

temperature, relative sea level, and coastal eutrophication on OM stabilization may yield the

potential to strongly reduce OM accumulation rates and increase wetland vulnerability to

accelerated SLR.

Our findings imply that particularly the vulnerability of organic systems might increase with

global change because in these systems soil volume is almost exclusively generated by the

preservation of OM. At the same time, however, mineral dominated systems, such as temperate

European salt marshes, experience large amounts of easily decomposable allochthonous-OM input

that relies on substantial stabilization in order to become sequestered (Middelburg et al., 1997;

Allen, 2000; Khan et al., 2015). Thus, future rates of C sequestration could be substantially reduced

in mineral dominated tidal wetland systems.

**Acknowledgements**

We thank Svenja Reents, Melike Yildiz, Anja Schrader, Detlef Böhm, Cailene Gunn, Johan Krol,

Marin van Regteren, Jacek Mazur, Ana Genua, Lluís Jornet, David Mateu, Sarah King, Shayne

Levoy, and Lyntana Brougham for help with field and lab work. This project was funded by the

Bauer-Hollmann Stiftung and the Rudolf und Helene Glaser Stiftung in the framework of the

INTERFACE project. The authors declare no conflict of interest.

**Author contributions**

PM, SN, KJ, and LMS-B designed the overall study. PM analyzed and interpreted the data. PM wrote the

initial version of the manuscript with regular comments and editing provided by LMS-B, TJM, and SN. PM,

LMS-B, TJM, GLC, TD, YK, AVdG, PE, CS, AD'A,CI, ML, UN, BJJ, AHB, SAY, DIM, ZY, and JW

designed and conducted the field studies in the respective sites and commented on an earlier version of the

manuscript.



**Figure captions**

**Figure 1** Overview map of study sites. *Notes:* See Table 1 for site details.

**Figure 2** (**a**) Decomposition rate (*k*) and (**b**) stabilization factor (*S*) versus mean atmospheric temperature during deployment period. Shown are results of linear regression analyses across and within elevation zones and organic and mineral soils.

**Figure 3** (**a** + **c**) Decomposition rate (*k)* and (**b** + **d**) stabilization factor (*S)* in high (orange) and low

(blue) elevated zones of tidal marsh and mangrove sites (compare Table 1). High and low elevated zones are characterized by distinct plant-species assemblages or by different stature of mangroves along the flooding gradient within each site. Shown are means ± SE for all sites (**a** + **b**) and for mineral, organic, marsh, and mangrove systems separately (**c** + **d**). P-values refer to results of paired t-tests (ns, *P* > 0.05; * *P* ≤ 0.05; ** *P* ≤ 0.01).

**Figure 4** Effects of marsh elevation (zone) and nutrient enrichment on both decomposition rate (*k*) and stabilization factor (*S*) in long-term enriched (filled bars) and reference areas (open bars) in the high marsh (*Spartina patens* zone) and low marsh (*Spartina alterniflora* zone) of the TIDE project site at the Plum Island Sound Estuary, Massachusetts, US. Shown are means ± SE and results of two-way ANOVAs and pairwise comparisons (Tukey's HSD test).





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





**Figure 1**

**Figure 1** Overview map of study sites. *Notes:* See Table 1 for site details.





**Figure 2**

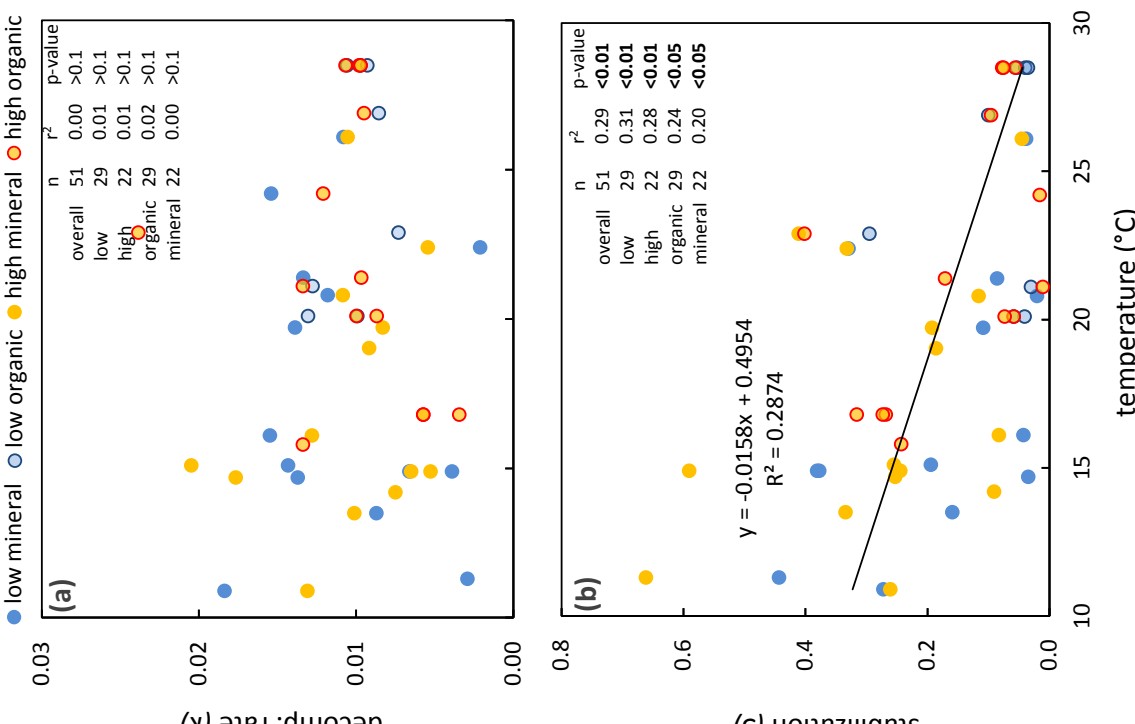

**Figure 2 (a)** Decomposition rate ($k$) and **(b)** stabilization factor ($S$) versus mean atmospheric temperature during deployment period. Shown are results of linear regression analyses across and within elevation zones and organic and mineral soils.





**Figure 3**

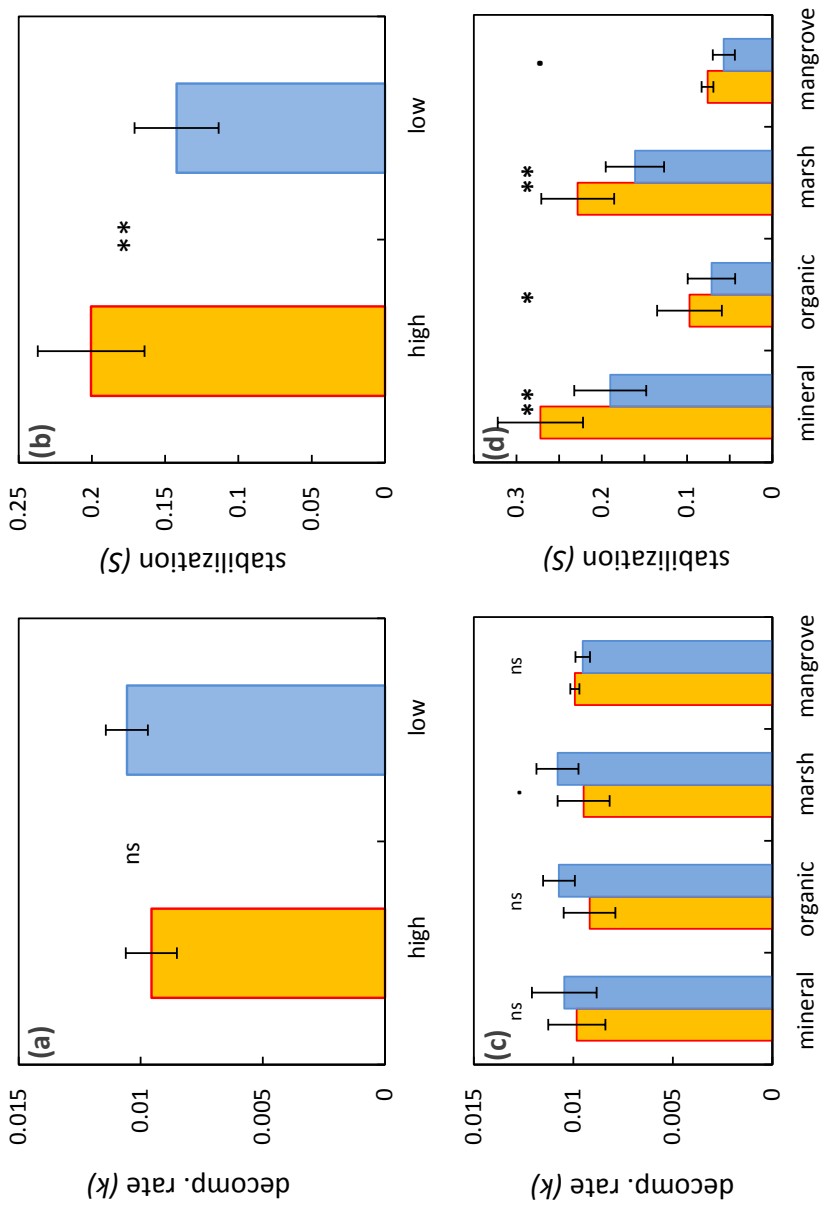

**Figure 3** (**a + c**) Decomposition rate ($k$) and (**b + d**) stabilization factor ($S$) in high (orange) and low (blue) elevated zones of tidal marsh and mangrove sites (compare Table 1). High and low elevated zones are characterized by distinct plant-species assemblages or by different stature of mangroves along the flooding gradient within each site. Shown are means ± SE for all sites (a + b) and for mineral, organic, marsh, and mangrove systems separately (c + d). P-values refer to results of paired t-tests (ns, $P > 0.05$; * $P \leq 0.05$; ** $P \leq 0.01$).



**Figure 4**

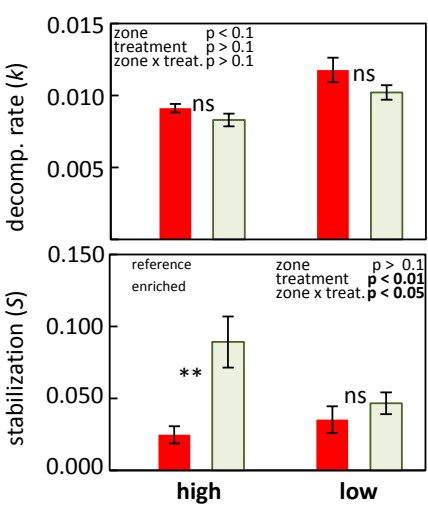

**Figure 4** Effects of marsh elevation (zone) and nutrient enrichment on both
decomposition rate (*k*) and stabilization factor (S) in long-term enriched (filled
bars) and reference areas (open bars) in the high marsh (Spartina patens zone) and
low marsh (Spartina alterniflora zone) of the TIDE project site at the Plum Island
Sound Estuary, Massachusetts, US. Shown are means ± SE and results of two-way
ANOVAs and pairwise comparisons (Tukey's HSD test).




**Table 1** Overview of study regions, site names, and site properties. Sites in which tea bags were deployed in zones of different elevation and flooding frequency are marked (x). Different salinity classes are indicated as 'S' (salt water), 'B' (brackish water), and 'F' (fresh water). Tidal amplitude (Ampl.) is given in meters.

| Region | Site name | Zonation | Salinity | Ampl. | Ecosystem | Soil[c] | Contact[site ref.] |
|---|---|---|---|---|---|---|---|
| *Europe* | | | | | | | |
| Germany | Dieksanderkoog | x | S | 3.0 | marsh | mineral | Mueller[1] |
|  | Sönke-Nissen-Koog | x | S | 3.4 | marsh | mineral | Mueller[1] |
|  | Spiekeroog | x | S | 2.0 | marsh | mineral | Dinter[2] |
| The Netherlands | Ameland | x | S | 2.3 | marsh | mineral | de Groot[3] |
|  | Noord-Friesland Buitendijks | x | S | 2.3 | marsh | mineral | Esselink[4] |
|  | Schiermonnikoog[b] | - | S | 2.3 | marsh | mineral | Smit[5] |
| Italy | Venice Lagoon | x | S | 0.5 | marsh | mineral | D'Alpaos[6] |
| Spain | Ebro Delta | x | F,B,S | <0.1 | marsh | organic | Ibáñez[7] |
| Poland | Mechelinskie Łąki[b] | - | B | <0.1 | marsh | organic | Lazarus |
| *North America* | | | | | | | |
| Canada, QC | Rimouski | x | S | 3.2 | marsh | mineral | Neumeier[8] |
| Canada, NB | Dipper Harbour | x | S | >6.0 | marsh | mineral | Chmura[9] |
| Maine, US | Long Marsh | - | F,B,S | 1.4 | marsh | organic | Johnson |
| Massachusetts, US | Laws Point | x | S | 2.9 | marsh | organic | Mozdzer[10] |
|  | TIDE project[a] | x | S | 2.9 | marsh | organic | Mozdzer[11] |
| Maryland, US | Patuxent | x | F | 0.7 | marsh | organic | Baldwin[12] |
|  | Rhode River | x | B | 0.2 | marsh | organic | Schile[13] |
| Virginia, US | Wachapreague | x | S | 0.6 | marsh | mineral | Schile[14] |
| California, US | Coon Island | x | B | 0.7 | marsh | mineral | Schile[14] |
|  | Rush Ranch | x | B | 0.7 | marsh | mineral | Schile[14] |
|  | China Camp | - | S | 0.7 | marsh | mineral | Schile[14] |
| *Central America* | | | | | | | |
| Belize | Twin Cays | x | S | 0.2 | mangrove | organic | Schile[15] |
| Panama | Isla Solarte | x | S | 0.3 | mangrove | organic | Schile[16] |
|  | Isla Cristóbal | x | S | 0.3 | mangrove | organic | Schile[16] |
|  | Isla Popa | x | S | 0.3 | mangrove | organic | Schile[16] |
| *South America* | | | | | | | |
| Argentina | Mar Chiquita[b] | x | B | 0.8 | marsh | mineral | Montemayor[17] |
| *Asia* | | | | | | | |
| China | Dongtan | x | S | 2.5 | marsh | mineral | Wu[18] |

(a) additional fertilization treatment was included, compare reference 10; (b) low retrieval rates of paired bags only allowed for calculation of site or zone averages (c) If unclear, soil type was judged as mineral at organic matter contents < 35% (Soil Survey Staff, 2014); Site references: (1) Nolte et al. (2013), (2) Flemming and Davis (1994), (3) Dijkema et al. (2010), (4) Chang et al. (2016), (5) Howison et al. (2015), (6) Roner et al. (2016), (7) Benito et al. (2014), (8) Neumeier and Cheng (2015), (9) (Chmura et al., 1997), (10) Morris et al. (2013), (11) Deegan et al. (2012), (12) Neff et al. (2009), (13) Langley and Megonigal (2010), (14) Vasey et al. (2012), (15) Mckee et al. (2007), (16) Lovelock et al. (2005), (17) Isacch et al. (2006), (18) Yang et al. (2017)





**Table 2** Hydrolysable (H) and mineral fractions of green tea (n = 5 batches) and rooibos tea (n = 3 batches) and C and N contents (n = 2 batches). Samples of each batch were analyzed as duplicates.

|  | Green Tea | | Rooibos Tea | |
|---|---|---|---|---|
|  | mean | SD | mean | SD |
| H [g g$^{-1}$] | 0.933 | 0.01 | 0.676 | 0.04 |
| Total C [%] | 47.9 | 2.8 | 50.1 | 0.7 |
| Total N [%] | 3.9 | 0.2 | 1.1 | 0.1 |
| Mineral fraction [%] | < 0.5 | | < 0.1 | |

**Table 3** Spearman's rank coefficients between the variables temperature (°C), latitude (°), tidal amplitude (m), salinity class, $k$, and $S$ (coefficients are bold typed at p ≤ 0.05) and comparisons of temperature, latitude, amplitude, k, and S between ecosystem types (mangrove vs. marsh) and soil types (mineral vs. organic) shown as means ± SE. Asterisks show results of Mann-Whitney U tests and denote significant differences as: p ≤ 0.05 = *,   p ≤ 0.01 = **, p ≤ 0.001 = ***, not significant = ns.

|  | temperature | latitude | amplitude | salinity | $S$ | $k$ |
|---|---|---|---|---|---|---|
| *Spearman's rank correlations* | | | | | | |
| temperature |  | **-0.71** | **-0.72** | -0.09 | **-0.49** | 0.02 |
| latitude | **-0.71** |  | **0.54** | 0.15 | **0.38** | 0.14 |
| amplitude | **-0.72** | **0.54** |  | **0.36** | 0.14 | 0.04 |
| salinity | -0.09 | 0.15 | **0.36** |  | 0.00 | -0.07 |
| $S$ | **-0.49** | **0.38** | 0.14 | 0.00 |  | **-0.48** |
| $k$ | 0.02 | 0.14 | 0.04 | -0.07 | **-0.48** |  |
| *group means ± SE* | | | | | | |
| **soil type** | *** | * | ** |  | * | ns |
| mineral | 17.3±0.8 | 40.2±4.2 | 2.23±0.40 |  | 0.23±0.03 | 0.010±0.001 |
| organic | 23.0±0.9 | 31.1±3.4 | 0.93±0.22 |  | 0.13±0.03 | 0.010±0.001 |
| **ecosystem** | *** | *** | ** |  | * | ns |
| marsh | 18.2±0.6 | 40.9±2.9 | 1.93±0.29 |  | 0.20±0.03 | 0.010±0.001 |
| mangrove | 28.1±0.3 | 11.4±1.3 | 0.28±0.02 |  | 0.07±0.01 | 0.010±0.000 |