# Peer review of "Global-change effects on early-stage decomposition processes in tidal wetlands – Implications from a global survey using standardized litter"

_Biogeosciences, 2017_

## Referee Comment (RC1) · Anonymous Referee #1 · 9 Jan 2018

General comments

This article deals with an important aspect of carbon's fate in coastal wetlands in relation to global changes and their impacts on these ecosystems. Indeed, wetlands are receiving a growing attention in the climate change debate in relation to their high capacity to sequester blue carbon. Ecosystems considered in this "global" scale study are mainly tidal marshes but some mangroves sites were counted in the selected sites. Authors are assessing OM degradation and transformation, as a proxy of Carbon sequestration using the TBI approach. Thus, authors claim that they provided indirect evidences that rising T° and Sea Level and eutrophication will impact the capacity of

tidal wetlands to sequester carbon. This work is worthwhile to publish although as authors cautioned, there are limits with the used method (obvious quality differences of Tea-bag OM with "real" plants) and also that they may have missed some influent factors that control OM degradation and sequestration.

Introduction was well thought and the methodology was clear however, some choices were not judicious in the context of this study and may need to be revaluated (see specific comments). The adding of TIDE experimental site was a very interesting. The discussion is well organised but it needs to be shortened.

Specific comments I am not a specialist of meta-analysis, therefore I will not comment on the validity or not of the numerical methods, but one thing is sure, analyses need always to rely on field knowledge even if results are "counterintuitive". The discussion is based on two characteristics (k , S) that are related to the quality and the fate of the litter-bags contents (here Tea-bags) which are strongly related to sedimentation dynamic and water velocity. In absence of a clear indication on how sediments (and OM) are behaving in each site, I am concerned about the amalgam in the same meta analysis different systems in term of hydrological functioning: Salt Marches vs. Mangroves, High tide vs. low tide (in salt marches). For instance, estuarine mangroves receive loads of sediments from rivers whereas Europeans salt marches in open Bays get sediments mainly from the oceans. One way to tackle this concern is to process the same calculations/test s/figures without adding the mangrove sites to the pool of data. Same thing can be done by considering the main origin of sediments (not to confound with OM), without impacted TIDE sites, river presence or not, water velocity, human activities. . . . These factors, of ecological importance, might be those missing to explain some global, or local, differences. If these data cannot be compiled they should at least be discussed.

---

## Short Comment (SC1) · 11 Jan 2018

Have you thought about the fact that you are the first ones that tested the TBI method in saltmarsh systems? In fact, you are only the second to test it outside the pure terrestrial system (see Sarneel et al 2017 for riparian systems). Since many people are interested in experiences of using TBI outside the terrestrial system, this is valuable study, as it expands the range over which it was used. However, although understandable for this study, presenting k and S based on new hydrolysable fractions does not help direct comparison with the values obtained in other studies. Could you provide the data following the standard method of keuskamp et al 2013 in an appendix? I further

wondered if you think that the salt water can have influenced what was the hydrolysable fraction?

---

## Referee Comment (RC2) · JA Keuskamp (Referee) · 21 Jan 2018

General comments

This paper discusses the control that the soil matrix exerts on the decomposition of organic matter in tidal wetlands. Their large carbon stocks and sensitivity to global change make this a highly relevant topic for scientists and policy makers alike. The paper is well-written and easy to read, while presenting novel data with important conclusions on the relation between decomposition and global change. The usage of a standardised method over a wide range of tidal systems allows for a generalisation to the global scale, making this paper relevant to the broad readership of Biogeosciences.

[Figure]

The explorative nature of the experiment also introduced some unavoidable methodological weaknesses. Many of the environmental parameters which are discussed in relation to decomposition are often strongly correlated with tidal regime (i.e. soil temperature, salinity, nutrient status, microbial biomass, and redox status), or latitude (i.e. nutrient limitation, vegetation type). In its current version, the manuscript does not always acknowledge the potentially spurious relation between these factors. While this does not invalidate the main conclusions I would recommend to consider non-causality more carefully when attributing effects to specific environmental parameters

The current description of the data-analysis does not describe how the authors have ascertained themselves that underlying assumptions of the statistical tests used were not violated. Where applicable, tests of heterogeneity, normality, and independence should be included, or other tests considered. For example a linear fitting is performed between k and S with temperature, without mentioning testing for residual patterns to uncover non-linearity. As the authors note the relation between decomposition and single parameters are often not linear (L221), in which case the result of a linear model is unreliable.

Lastly, I would like to add that the strength if the TBI lays in its standardisation. I would therefore recommend to mention the S/k calculated with the standard approach alongside with the re-scaled values calculated with the more aggressive extraction method. This would allow for easy comparison with other data such as the TBI-values from mangroves mentioned in the methods paper. See also below.

specific comments

L79 and L83-L84 seem largely redundant to me

L85-L86 'OM decomposition' is somewhat ambitious as it is not clear whether this refers to decomposition rate (k) or extend (S), please revise.

L117 Although this should have been more explicit in the TBI method paper (Keuskamp

et al, 2013), the k estimated by TBI is not exactly equivalent to the classical litter bag experiment as it describes the decomposition rate of the hydrolysable fraction and is not calculated over the entire mass. We have therefore adapted k1 to indicate that this is the k of the most labile fraction, as opposed to k2 which refers to the decomposition rate of the recalcitrant fraction. to the To avoid confusion this should be made explicit here.

L120 The recalcitrant fraction is also decomposable, albeit a lot slower

L127 ' thereby improving our process-level understanding on how global warming affects carbon turnover' Not sure what this means exactly

L137 I am somewhat surprised that the oxidation of organic matter would be limited by the supply of SO4 in brackish tidal wetlands. Wouldn't the constant flushing with water replenish SO4 to saturating levels in brackish/salt water systems?

L154 '(i.e. dwarf vs. fringe phenotypes)' Aren't these also Rhizophora vs Avicennia? In that case phenotypes would not be the appropriate description. These mangroves belong to different genera, each with their own properties (soil oxygenation, phenolic compound production, N-content) that are known to influence decomposition.

L154 'Relative elevation' as relative to what? mean lower tide, mean mean tide? please specify

L169-170 Decomposition rates depend on soil temperature rather than on air temperature. Others have shown (e.g Piccolo et al. 1993, Reckless et al. 2011) that in tidal wetlands, the soil temperature is strongly determined by inundation regime in which case the accuweather temperature are not an accurate reflection of the decomposition environment. Moreover, inundation regime and temperature effects would be confounded. Could it be shown accuweather estimated temperatures vs measured temperatures so that the reader can see for themselves whether the accuweather approximation suffices?

L176 Pepsico, to my knowledge the bags are produced by Lipton, which is a Unilever brand.

L180 Were the reference bags dried at 70oC prior to mass determination?

L198-L200 It could well be that the method described is a more accurate operational-isation of the labile (non-hydrolysable) fraction. Redefining the labile fraction and the consequential shift in S, and rescaling of k, may however lead to misunderstandings when the results of this study are used in comparisons with other TBI experiments. I would therefore suggest to provide the TBI S/k values calculated according to protocol alongside the obtained S/k values obtained by the revised protocol.

L220-L250 Would you be able to indicate whether potential violations of the assumptions underlying the statistical tests were assessed? For example, were the residuals of the ANOVA procedure tested for normality / homogeneity of variance?

L250 It is critical to this conclusion that air temperature is a good proxy of soil temperature (see earlier remark). The interaction between temperature effect and tidal position reinforces the suspicion that this is not the case.

L314 As also noted in L313, the absence of a temperature effect is very unusual. Could the authors rule out the possibility that this is due to a mismatch between soil and air temperature?

L332 I would recommend discussing potential confounding of temperature effects with other changes in decomposition matrix(e.g. nutrient availability, redox status, vegetation, salinity). With respect to k, such reservations are made in L323/L329, but are absent here.

L351 Can this be generalised to continuously submerged parts of the soil? The TBI is at a relatively low depth, where tidal pumping may cause increased influx of oxygen during tidal subsidence. Especially in tannin-rich mangrove systems, temporal oxygenation may make a large difference by allowing breakdown of phenolic compoounds (see also

Freeman et al, 2001)

L445 In mangrove TBI experiments that I have conducted S values have always been positive, and I am somewhat puzzled by the large difference. Negative S values could also be caused by loss of recalcitrant particles as I have observed when using teabags in open water. Did you have any indications that this has taken place here?

Technical corrections L74 Earth? Not sure if this should be with a capital E L77 Separate SRL from citations L94-98 This sentence is very hard to read. Split. L346 add 'in' before 'tidal wetlands'

References

M.C. Piccolo, G.M.E. Perillo, G.R. Daborn, Soil Temperature Variations on a Tidal Flat in Minas Basin, Bay of Fundy, Canada, Estuarine, Coastal and Shelf Science, Volume 36, Issue 4, 1993, Pages 345-357, ISSN 0272-7714, https://doi.org/10.1006/ecss.1993.1021.

Klaus Ricklefs, Klaus Heinrich Vanselow, Analysis of temperature variability and determination of apparent thermal diffusivity in sandy intertidal sediments at the German North Sea coast, Estuarine, Coastal and Shelf Science, Volume 108, 2012, Pages 7-15, ISSN 0272-7714, https://doi.org/10.1016/j.ecss.2011.09.015.

Freeman, C., Ostle, N., & Kang, H. (2001). An enzymic "latch" on a global carbon store. Nature, 409(6817), 149–149. http://doi.org 10.1038/35051650

---

## Referee Comment (RC3) · Anonymous Referee #3 · 6 Feb 2018

Mueller et al. conducted decomposition experiments using tea bags based on a standardized approach developed by Keuskamp et al. (2013), across different marsh and mangrove sites in order to cover a gradient in temperature, indundation regime, etc. While such cross-ecosystem studies have a high potential, I feel the impact of this dataset in terms of new insights is relatively limited. The dataset can be published but I feel the impact of the conclusions should be toned down somewhat – the manuscript does not really deliver what the title suggests. The datasset should be publishable, but it needs a more critical discussion and should provide the readers with a more complete overview of the caveats and assumptions used in the TBI approach, so that the readers can better assess what can and cannot be deduced from these data.

[Figure]

My main point is that the TBI index – both the original and the modified protocol suggested here – has plenty of limitations and it remains an operationally defined procedure, with several assumptions that are open to discussion. In addition, we are not looking at mineralization of in situ produced material hence some interactive effects will be missed in this approach; results should not be over-interpreted or generalized.

Specific suggestions -L55: "stabilization was 29% lower": this does not mean much if you do not define stabilization here, it can be interpreted in different ways. For me this remains a somewhat problematic proxy (see further comments).

-L60-61: data from the eutrophication experiment: would not extrapolate this to 'high sensitivity to global change'. Eutrophication will also affect the nutrient content of locally produced biomass, this aspect is not taken into account when standardized material is used in the experiments.

-L90-95: an important caveat here is that you only study the decomposition of one type of source material (well, in two versions), but not other sources that contribute to the OM pool e.g. marine or other aquatic inputs into the intertidal system.

-section 2.2: it is important for the readers not familiar with the Keuskamp et al. paper to re-iterate and stress the assumptions on which this approach is based, e.g. that $k2$ (decomposition constant of the non-labile fraction) is assumed to be 0, and that S is assumed to be similar for both types of tea. I still find this major shortcomings- we know the first assumption not to be valid, and I have not seen strong arguments to support the second assumption. The main reason to make these assumptions is to allow to estimate k and S using only one time point of measurements instead of having to measure at different points in time. These aspects deserve to be mentioned explicitly and the limitations of the approach should be discussed more critically. -What is the added value of this approach compared to simply measuring the decay of the biomass over a limited number of time steps, and using a more realistic decay model ?

-L212-214: provide the data from Keuskamp et al. as well, we cannot compare or

assess how much higher your data are.

-discussion L 427-434: This is somewhat problematic also. It demonstrates the disadvantages of using these operationally defined indices; to which extent is this caused by the assumption that S is identical for the two types of substrate ? Secondly, keep in mind that anaerobic decomposition processes are important in tidal wetlands, and can occur at high rates (similar order of magnitude as aerobic decomposition) up to substantial depths.

---

## Author Comment (AC1) · 26 Feb 2018

We thank all three reviewers as well as the author of the Interactive Comment for their constructive comments and suggestions. In the pages below we respond to each of these separately. Unless stated otherwise, we intend to incorporate these changes into a final version of the manuscript.

In the following, reviewer comments are shown in in black, author responses in green font.

**REVIEWER 1, anonymous**

**General comments**
This article deals with an important aspect of carbon's fate in coastal wetlands in relation to global changes and their impacts on these ecosystems. Indeed, wetlands are receiving a growing attention in the climate change debate in relation to their high capacity to sequester blue carbon. Ecosystems considered in this "global" scale study are mainly tidal marches but some mangroves sites were counted in the selected sites. Authors are assessing OM degradation and transformation, as a proxy of Carbon sequestration using the TBI approach. Thus, authors claim that they provided indirect evidences that rising Temperature and Sea Level and eutrophication will impact the capacity of tidal wetlands to sequester carbon. This work is worthwhile to publish although as authors cautioned, there are limits with the used method (obvious quality differences of Tea-bag OM with "real" plants) and also that they may have missed some influent factors that control OM degradation and sequestration. Introduction was well thought and the methodology was clear however, some choices were not judicious in the context of this study and may need to be revaluated (see specific comments). The adding of TIDE experimental site was a very interesting. The discussion is well organised but it needs to be shortened.

We thank the reviewer for his positive feedback concerning the overall structure of our study! As requested, we will streamline the discussion where appropriate; however, the total length of the discussion will also depend on the required additions to the section on Methodological considerations (4.4).

**Specific comments**
I am not a specialist of meta-analysis, therefore I will not comment on the validity or not of the numerical methods, but one thing is sure, analyses need always to rely on field knowledge even if results are "counterintuitive". The discussion is based on two characteristics (k, S) that are related to the quality and the fate of the litter-bags contents (here Tea-bags) which are strongly related to sedimentation dynamic and water velocity. In absence of a clear indication on how sediments (and OM) are behaving in each site, I am concerned about the amalgam in the same meta-analysis different systems in term of hydrological functioning: Salt Marches vs. Mangroves, High tide vs. low tide (in salt marches). For instance, estuarine mangroves receive loads of sediments from rivers whereas Europeans salt marches in open Bays get sediments mainly from the oceans. One way to tackle this concern is to process the same calculations/test s/figures without adding the mangrove sites to the pool of data. Same thing can be done by considering the main origin of sediments (not to confound with OM), without impacted TIDE

sites, river presence or not, water velocity, human activities: … . These factors, of ecological importance, might be those missing to explain some global, or local, differences. If these data cannot be compiled they should at least be discussed.

We agree with the reviewer that the different systems we compiled in a single meta-analysis are characterized by potentially important differences in both sediment load and origin. We did not explicitly assess sediment loads of our study sites. However, by distinguishing between minerogenic and organogenic systems (i.e. sediment rich vs. sediment poor systems) in our analyses, we are confident to have already captured the relative importance of sediment load on our response variables. Please note that this categorical factor did not show up to be important in our classification- & regression-tree analysis (CRTA). Furthermore, our two most important findings (i.e. S decreases with temperature; S is lower in low vs. high elevated zones) are consistent within both minerogenic and organogenic systems.

We indirectly also addressed sediment origin (riverine vs. marine) by including both estuarine and coastal systems in our study. Specifically, we tested for effects of salinity class (fresh, brackish, salt) on our response variables, with fresh systems far up in the estuary experiencing the lowest marine influence and salt-water systems experiencing the highest marine influence. If sediment origin (riverine vs. marine) had an important influence on our response variables, this should have been reflected in our meta-analyses. That being said, salinity of floodwater and sediment origin can of course not easily be separated in an observational study. Concerning the reviewer's remark on the sediment origin of our mangrove sites, it needs to be noted that those were not estuarine (as assumed by the reviewer) but coastal systems in the present study.

We agree with the reviewer that factors other than those assessed in this study (i.e. human activities, river presence) might have been influential and could have masked expected results (i.e missing temp effect on k). In accordance with the reviewer's suggestion, we will elaborate on this in the discussion. Specifically, this will add to the section L325-330.

**REVIEWER 2, Dr J. Keuskamp**

**General comments**
This paper discusses the control that the soil matrix exerts on the decomposition of organic matter in tidal wetlands. Their large carbon stocks and sensitivity to global change make this a highly relevant topic for scientists and policy makers alike. The paper is well-written and easy to read, while presenting novel data with important conclusions on the relation between decomposition and global change. The usage of a standardised method over a wide range of tidal systems allows for a generalisation to the global scale, making this paper relevant to the broad readership of Biogeosciences. The explorative nature of the experiment also introduced some unavoidable methodological weaknesses. Many of the environmental parameters which are discussed in relation to decomposition are often strongly correlated with tidal regime (i.e. soil temperature, salinity, nutrient status, microbial biomass, and redox status), or latitude (i.e. nutrient limitation, vegetation type). In its current version, the manuscript does not always acknowledge the potentially spurious relation between these factors. While this does not

invalidate the main conclusions I would recommend to consider non-causality more carefully when attributing effects to specific environmental parameters.

We thank Dr Keuskamp for his constructive comments on our study! We agree that correlations between the assessed environmental parameters should be carefully considered in the interpretation of our results. Specifically, the reviewer is concerned about interactions with the parameters tidal regime and latitude.

In terms of describing the tidal regime, we assessed tidal amplitude and, by comparing high and low elevated zones within sites, a relative measure for flooding frequency (i.e. low zones more frequently flooded than high zones). Although, tidal amplitude did not affect k and S (Table 3), it showed up as a potentially important predictor in our CRTA, probably because of its strong correlation with other parameters. In L289-292 we addressed this: "[…] this result needs to be interpreted cautiously […] effects of tidal amplitude are not independent from other factors because strong correlations existed with ecosystem and soil type, temperature, and latitude (Table 3)." This is the Results section, though, and we will put more emphasize on this in the discussion, adding to L325-330 and to L346.

In terms of flooding frequency (high vs. low elevated zones), I am confident that we already discussed a number of potentially relevant interactions that were mentioned by the reviewer: redox -> L360-362; salinity -> L363-365; nutrient status -> L371-376. The reviewer makes an important point by mentioning soil temp interactions with tidal regime. We will address this point further below where soil vs air temp differences are discussed.

Changes in nutrient status/limitation and vegetation type with latitude are indeed relevant for the interpretation of the temperature effects on S and k. We will extend the following section of our discussion: "We therefore conclude that other parameters exerted overriding influence on k, mainly masking temperature effects and have not been capture by our experimental design. For instance […]" (L322-330). The same discussion is yet missing for S (as noticed by the reviewer below). This will be addressed accordingly, adding to line 346.

The current description of the data-analysis does not describe how the authors have ascertained themselves that underlying assumptions of the statistical tests used were not violated. Where applicable, tests of heterogeneity, normality, and independence should be included, or other tests considered.

The reviewer is correct, we will give more detailed information on this in the statistics section as follows (new -> highlighted yellow):

"Two-tailed paired t-tests were used to test for effects of relative elevation as proxy for relative sea level on *k* and *S* (Fig. 3). Subsequent one-tailed paired t-tests were conducted to test for the same effect within mineral, organic, marsh, and mangrove systems separately. The absence of outliers and normal distribution of the difference in the independent variable (as assessed visually) assured robustness of paired t-tests.

In 21 of our 22 sites where tea bags were deployed in both high and low elevation zones, replication was sufficient to conduct one-way ANOVA to test for differences in *k* and *S* between zones for each site separately (Fig. S2). Normal distribution of residuals was assessed visually, Levene's test was used to test for homogeneity of variance, and data were log-transformed if

assumptions were not met. Mann-Whitney U tests were conducted as a non-parametric alternative when log-transformed data did not meet ANOVA assumptions (Fig. S2). We tested for effects of nutrient enrichment on *k* and *S* in the data from the TIDE site (Massachusetts, US) using two-way ANOVA with enrichment treatment and marsh zone as predictors. When Levene's test indicated heterogeneous variance (true for *k*), data were log-transformed, which stabilized variance. Normal distribution of residuals was assessed visually."

For example a linear fitting is performed between k and S with temperature, without mentioning testing for residual patterns to uncover non-linearity. As the authors note the relation between decomposition and single parameters are often not linear (L221), in which case the result of a linear model is unreliable.

The reviewer is correct. A linear effect of temp is not expected. The intention for showing the linear fit was only to better illustrate the significant temp effect on S (as tested/identified with non-parametric Spearman correlation). However, we should not have used linear regression as an additional hypothesis test, and we will only use Spearman's correlation for this in the revised version. Yet, to better illustrate the temp effects, we will still present scatterplots (as in Fig. 2) and will use curve fitting to illustrate significant temp effects. Indeed, the model with both highest R2 and lowest standard error of estimate describing the significant temp effect on S is not linear but a logarithmic.

Lastly, I would like to add that the strength if the TBI lays in its standardisation. I would therefore recommend to mention the S/k calculated with the standard approach alongside with the re-scaled values calculated with the more aggressive extraction method. This would allow for easy comparison with other data such as the TBI-values from mangroves mentioned in the methods paper. See also below.

We agree with the reviewer. The same point has been raised by Dr Sarneel in an interactive comment (below). We have prepared a table with all site x zone values for k and S, giving both the original TBI-values and the modified. This table will be put in the Supporting Information and referenced in the manuscript.

**Specific comments**
L79 and L83-L84 seem largely redundant to me

Will delete 79-80 accordingly

L85-L86 'OM decomposition' is somewhat ambitious as it is not clear whether this refers to decomposition rate (k) or extend (S), please revise.

The sentence will be changed to "Consequently, global changes that might decrease OM preservation in tidal wetland soils not only affect carbon sequestration, but also decrease ecosystem stability against SLR." Obviously, preservation is also affected by decomposition rate and stabilization; however, we do not intend to specify the processes at this stage of the Introduction, but do this further down in the text (i.e. 107-113).

L117 Although this should have been more explicit in the TBI method paper (Keuskamp et al, 2013), the k estimated by TBI is not exactly equivalent to the classical litter bag experiment as it describes the decomposition rate of the hydrolysable fraction and is not calculated over the entire mass. We have therefore adapted k1 to indicate that this is the k of the most labile fraction, as opposed to k2 which refers to the decomposition rate of the recalcitrant fraction. to the To avoid confusion this should be made explicit here.

We will avoid reference to classical litter bag experiments here and instead make the meaning of *k* clearer in the respective section of the Methods.

L120 The recalcitrant fraction is also decomposable, albeit a lot slower

Good catch, this was poor wording of course. Will be changed to "rapidly decomposable".

L127 ' thereby improving our process-level understanding on how global warming affects carbon turnover' Not sure what this means exactly

We will delete "process-level".

L137 I am somewhat surprised that the oxidation of organic matter would be limited by the supply of SO4 in brackish tidal wetlands. Wouldn't the constant flushing with water replenish SO4 to saturating levels in brackish/salt water systems?

Well, it probably depends on how much seawater input the brackish system experiences. Anyhow, we will delete L136-139 but keep the lines above adding Craft, 2007 and Weston et al., 2006 to provide the reader with more context: "Additionally, by expanding our study to include fresh and brackish sites, we anticipated to capture the effects of salt-water intrusion into brackish and fresh systems, which is likely to affect decomposition processes in tidal wetlands (Craft, 2007; Morrissey et al., 2014; Weston et al., 2006)."

L154 '(i.e. dwarf vs. fringe phenotypes)' Aren't these also Rhizophora vs Avicennia? In that case phenotypes would not be the appropriate description. These mangroves belong to different genera, each with their own properties (soil oxygenation, phenolic compound production, N-content) that are known to influence decomposition.

In most cases you would assume so, but here both fringe and dwarf are indeed Rhizophora with very few individuals of Avicennia: Please see: Mckee et al. (2007) Global Ecology and Biogeography, **16**, 545–556; Lovelock et al. (2005) Caribbean Journal of Science, Vol. 41, No. 3, 456-464, 2005

L154 'Relative elevation' as relative to what? mean lower tide, mean mean tide? Please specify

We will add the following (yellow highlight) to make clear that it refers to the previous sentence:

"At 22 sites, we compared high and low elevated zones, which were characterized by distinct plant species compositions (i.e. different communities in high vs. mid vs. low marshes) or by different stature of mangroves (i.e. dwarf vs. fringe phenotypes). We used relative elevation (i.e. high vs. low elevated zone) as a site-specific proxy for relative sea level."

L169-170 Decomposition rates depend on soil temperature rather than on air temperature. Others have shown (e.g Piccolo et al. 1993, Reckless et al. 2011) that in tidal wetlands, the soil temperature is strongly determined by inundation regime in which case the accuweather temperature are not an accurate reflection of the decomposition environment. Moreover, inundation regime and temperature effects would be confounded. Could it be shown accuweather estimated temperatures vs measured temperatures so that the reader can see for themselves whether the accuweather approximation suffices?

Dr Keuskamp brings a valid point here that indeed needs more consideration. Air temperature would obviously diverge from soil temperature depending on factors such as canopy shading or tidal regime and water temperature. As a consequence, air temperature can only approximate the temperature conditions of the actual decomposition environment. However, considering that we stretch a temp gradient of approx. 20°C, we are confident that this would also translate into a profound soil-temperature gradient across our study sites.

The two studies mentioned by the reviewer, Piccolo et al. 1993 and Ricklefs et al. 2012, present data for un-vegetated tidal flat systems. For marsh systems, we would rather refer to Kirwan et al. (2014). The authors show, that in marshes along the well-studied latitudinal gradient of the US East coast (and we do share a number of sites), soil temp and air temp are highly correlated, while the relationship between soil temp and water temp is far weaker (Kirwan et al. 2014, *Temperature sensitivity of organic-matter decay in tidal marshes*; biogeosciences: **Fig. 2a**).

In our study sites, we did not continuously measure soil temp over the 3 months of deployment, and thus it is difficult to assess how well soil and air temp were correlated in this study. However, in several of our sites, soil temp was assessed at the time point of insertion and retrieval of bags. We plotted these data against the mean air temp of the day as acquired from the *accuweather* service in the Figure below:

[Figure]

We see that generally air temp is a good proxy for soil temp across sites. Yet, there is considerable variability in soil temp not explained by air temp, which would result from the fact that soil temp was assessed in one time point as opposed to mean air temp of a single day and of course from other factors, such as distance of weather station from site, shading, influence of water temp etc..

We agree with the reviewer that this needs to be stated and discussed in the manuscript. In the Methods we will add accordingly: ". It needs to be noted here, that top-soil temperature would differ from air temperature depending on factors such as canopy shading or tidal regime and water temperature. As a consequence, air temperature can only approximate the temperature conditions of the actual decomposition environment (Fig. SX)." Furthermore, we will add this point to the discussion on the missing temp and elevation effects on k, and we will present this figure in the Supporting Information.

Lastly, we want to stress a related point here: "low" and "high" in the figure legend refer to the low and high elevated zones within the systems. A paired t-test comparing the difference of air temp and soil temp between the paired high and low elevated zones indicates no significant effect of zone (p = 0.563). This shows that differences between air and soil temp were not consistently more pronounced in either the low or the high elevated zones. Additionally, soil temp was not significantly affected by zone (p = 0.342). One of our main findings, that S is consistently lower in low vs. high zones, is consequently not temperature affected (i.e. S was significantly reduced in 15 of 21 sites, and the opposite was observed in none of the sites (Fig. S2)).

L176 Pepsico, to my knowledge the bags are produced by Lipton, which is a Unilever brand.

Unilever belongs to PepsiCo, but of course the tea is produced by Unilever. Will delete PepsiCO in order to avoid confusion.

L180 Were the reference bags dried at 70oC prior to mass determination?

This may be a misunderstanding: reference bags were used to determine a mean value of the empty nylon bag itself without contents. I do not know if that one has always been dried. Anyhow, empty bag weights were very consistent among labs.

However, initial tea-content weights showed large variability across the involved labs. I also noticed that some labs, after drying at 70°C, used desiccators, in which the material could cool down without sucking moisture, before weighing and some didn't. I therefore assessed if potential moisture differences of the initial tea material or differences in the amount of the initial material could have affected S or k. However, there was no relationship between green initial weight and S (r2=0.0003; p=0.936) and no between rooibos initial and k (r2=0.005; p=0.728).

L198-L200 It could well be that the method described is a more accurate operationalisation of the labile (non-hydrolysable) fraction. Redefining the labile fraction and the consequential shift in S, and rescaling of k, may however lead to misunderstandings when the results of this study are used in comparisons with other TBI experiments. I would therefore suggest to provide the TBI S/k values calculated according to protocol alongside the obtained S/k values obtained by the revised protocol.

We agree. Will add a table with the original TBI values accordingly (see comment further up).

L220-L250 Would you be able to indicate whether potential violations of the assumptions underlying the statistical tests were assessed? For example, were the residuals of the ANOVA procedure tested for normality / homogeneity of variance?

This was indeed missing. We added these missing details to the Methods, see comment further up.

L250 It is critical to this conclusion that air temperature is a good proxy of soil temperature (see earlier remark). The interaction between temperature effect and tidal position reinforces the suspicion that this is not the case.

We agree with the first half of this remark (see addressed further up), but not with the second. That is, there is no clear interaction between tidal position and temperature: Temperature seems to affect k in mesotidal systems (tidal amp >2.1m) with k higher in sites with temp >14.5°C; however, this apparent temp effect is inconsistent within this group of mesotidal systems. That is, sites with temp >18.2°C show lower k than those sites with temp <18.1°C.

L314 As also noted in L313, the absence of a temperature effect is very unusual. Could the authors rule out the possibility that this is due to a mismatch between soil and air temperature?

We stretch large gradients of approx. 20°C for both soil and air temp, and there is not even the slightest tendency of a temp effect on k (Spearman's rank coefficient = 0.02; Figure 2), while S is strongly affected. It therefore seems that that other factors exert overriding control over k and

more strongly mask temp effects on k than on S. Yet, we agree on the need to discuss the methodological inaccuracy in determining temp of the decomposition environment in the discussion, and we will address this adding to L325-330 and L346.

We want to stress a related point here concerning the missing temp effect on k: In order to address remarks by Reviewer 3, we took a separate look at the data of the US East coast latitudinal gradient along which previous studies have shown clear temp effects on decomposition processes microbial biomass (Blum et al. 2004; Kirwan et al. 2014; Mozdzer et al. 2014). Species composition of these marshes is quite constrained (i.e. Spartina alterniflora dominated) reducing confounding effects induced by differences in vegetation. Along this gradient, we clearly see an increase in S and also the expected decrease in k, although temp explains more variability for S. We intend to add this figure (shown below) to the manuscript in order to illustrate that temp effects on k can be identified on the regional scale, but not on the global scale with more confounding factors.

The effect of temp on k at the regional scale but the missing effect at global scale is also in agreement with the just recently published article on *Early stage litter decomposition across biomes* by Ika Djukic and others. Although they did not assess specifically k and S in their study using the TBI tea materials, they simply assessed mass loss of the two materials: Across biomes, climate (temp and precipitation) had no effect on break down; however, within biomes the effect was strong.

L332 I would recommend discussing potential confounding of temperature effects with other changes in decomposition matrix (e.g. nutrient availability, redox status, vegetation, salinity). With respect to k, such reservations are made in L323/L329, but are absent here.

We agree with the reviewer and will add similar considerations for the discussion on temp effects on S (L346).

L351 Can this be generalised to continuously submerged parts of the soil? The TBI is at a relatively low depth, where tidal pumping may cause increased influx of oxygen during tidal subsidence. Especially in tannin-rich mangrove systems, temporal oxygenation may make a large difference by allowing breakdown of phenolic compounds (see also Freeman et al, 2001)

We agree. The depth component needs to be stressed here briefly as already done in the section on Methodological considerations: "First, we demonstrate that *S* is indeed reduced in low vs. high elevated zones across our study sites, indicating that redox conditions in the top soil of tidal wetlands are at least often not low enough to hamper decomposition processes." (L432-433). We will add this point to L351.

L445 In mangrove TBI experiments that I have conducted S values have always been positive, and I am somewhat puzzled by the large difference. Negative S values could also be caused by loss of recalcitrant particles as I have observed when using teabags in open water. Did you have any indications that this has taken place here?

We were puzzled as well when realizing that so many values were lower than they should be. Indeed, the FL mangrove values you report in Keuskamp et al 2013 are considerably higher. That's also when I decided to check whether the quality of the material had changed. No, I am not aware of loss of particles from the bags in situ. Comparing our results to those reported in Djukic et al. (2018), it becomes clear that negative S values occur less frequently across terrestrial systems, however, are not negligible either.

Technical corrections L74 Earth? Not sure if this should be with a capital E
L77 Separate SRL from citations
L94-98 This sentence is very hard to read. Split.
L346 add 'in' before 'tidal wetlands'

Thanks, we will address these technical corrections.

**REVIEWER 3, anonymous**

Mueller et al. conducted decomposition experiments using tea bags based on a standardized approach developed by Keuskamp et al. (2013), across different marsh and mangrove sites in order to cover a gradient in temperature, inundation regime, etc. While such cross-ecosystem studies have a high potential, I feel the impact of this dataset in terms of new insights is relatively limited. The dataset can be published but I feel the impact of the conclusions should be toned down somewhat – the manuscript does not really deliver what the title suggests. The dataset should be publishable, but it needs a more critical discussion and should provide the readers with a more complete overview of the caveats and assumptions used in the TBI approach, so that the readers can better assess what can and cannot be deduced from these data. My main point is that the TBI index – both the original and the modified protocol suggested here – has plenty of limitations and it remains an operationally defined procedure, with several assumptions that are open to discussion. In addition, we are not looking at mineralization of in situ produced material hence some interactive effects will be missed in this approach; results should not be over-interpreted or generalized.

We thank the reviewer for his critical and constructive feedback on our work! We are willing to make the necessary changes to manuscript, especially to the discussion part (4.4 Methodological considerations; 4.5 Implications), in order to provide the reader with a more complete overview of the assumptions involved with the TBI approach. We will furthermore specify where conclusions/implications of our results are limited by the approach. We have provided more detailed responses below regarding the specific comments raised by the reviewer.

**Specific suggestions**

L55: "stabilization was 29% lower": this does not mean much if you do not define stabilization here, it can be interpreted in different ways. For me this remains a somewhat problematic proxy (see further comments).

We agree with the reviewers concern and will briefly specify the TBI parameters in the abstract. The second part of the comment will be addressed further below.

L60-61: data from the eutrophication experiment: would not extrapolate this to 'high sensitivity to global change'. Eutrophication will also affect the nutrient content of locally produced biomass, this aspect is not taken into account when standardized material is used in the experiments.

The reviewer is of course correct to state that with eutrophication, also the quality of the biomass produced in the system would change with potentially important consequences for the decay process. Thus, interpretation of the results obtained with standardized litter need to be conducted cautiously. Please note that we already tried to do this in our discussion→

L412-422: "The quality of OM (i.e. its chemical composition) is a key parameter affecting its decomposition. As the quality of the TBI materials differ from that of wetland plant litters, we did not expect to capture precise and absolute values for wetland litter break down in this study. Instead, we used the Tea-Bag Index to characterize the decomposition environment by obtaining a measure for the potential to decompose and stabilize the deployed standardized material. Standardized approaches like this, or also the cotton-strip assay (e.g. Latter and Walton, 1988), are useful to separate the effects of environmental factors other than OM quality on decomposition processes and to assess their relative importance. Otherwise, complex interaction effects of the abiotic environment and OM quality make it difficult to predict the relevance of certain environmental factors for decomposition processes, potentially masking the effects of important global-change drivers (reviewed in Prescott, 2010)."

We agree with the reviewer that this sentence needs to be toned down in the abstract, because there is no space for further elaboration on the assumptions and methodological considerations. Will change sentence to: "Additionally, data from the long-term ecological research site in Massachusetts, US show a pronounced reduction in the stabilization of the standardized OM by >70% in response to simulated coastal eutrophication, confirming the potentially high sensitivity of OM stabilization to global change." Please note that "confirming the potentially high sensitivity […] to global change" refers to the findings reported prior (i.e. decrease of S with temperature and lower S in low vs. high elevated zones).

L90-95: an important caveat here is that you only study the decomposition of one type of source material (well, in two versions), but not other sources that contribute to the OM pool e.g. marine or other aquatic inputs into the intertidal system.

We agree; this is important for the interpretation of our results. However, conventional litter bag experiments are also restricted in their choice of material; here actually lies an advantage of the standardized approach, although we acknowledge that the quality of the TBI materials is

obviously closer to that of wetland plant litter than to the marine derived, labile allochthonous organic input a tidal wetland receives. We will add this point to section 4.4 (L411-422) and make sure that Implications (4.5) address this accordingly.

section 2.2: it is important for the readers not familiar with the Keuskamp et al. paper to re-iterate and stress the assumptions on which this approach is based, e.g. that k2 (decomposition constant of the non-labile fraction) is assumed to be 0, and that S is assumed to be similar for both types of tea. I still find this major shortcomings- we know the first assumption not to be valid, and I have not seen strong arguments to support the second assumption. The main reason to make these assumptions is to allow to estimate k and S using only one time point of measurements instead of having to measure at different points in time. These aspects deserve to be mentioned explicitly and the limitations of the approach should be discussed more critically. -What is the added value of this approach compared to simply measuring the decay of the biomass over a limited number of time steps, and using a more realistic decay model?

Please note that that Keuskamp et al. (2013) show that the TBI is quite robust against deviations from the assumption that S is the same for the two materials. The assumption that k2 could be considerably higher than 0 during 3 months of deployment has already been questioned by us in 4.4 (Methodological considerations) → L438-440. However, the reviewer brings important points here: In accordance, we will elaborate on the description of the TBI calculations in the respective section of the Methods (2.2). Further, we will discuss the resulting limitations in our section on the Methodological considerations (4.4).

The advantage of this approach over a longer-term litter experiment is the time efficiency that allowed us to assess decomposition in a large number of sites during the same growth season and find enough collaborators capable to contribute with their work. Obviously, as lined out by the inventers of the method (Keuskamp et al. 2013), the TBI can't substitute the precision of classic litter bag methods, but it considerably reduces the effort necessary to fingerprint local decomposition. It is a trait-off between precision and effort that helps gathering decomposition data across ecosystems and biomes.

In order to demonstrate the usefulness of the method and its comparability to other methods assessing decomposition processes tidal wetlands, we will separately present our data on k and S for the sites along the US East coast latitudinal gradient, along which previous studies have shown clear temperature and latitudinal effects on decomposition processes. For instance, Kirwan et al. (2014; Biogeosciences) demonstrated a strong increase in litter decay and a strong decrease in cotton tensile strength with both temperature and latitude, and Mozdzer et al. (2014; Ecology) showed a marked decrease in sulfate reduction (dominant anaerobic decomposition process in salt marshes) with latitude along this transect →

[Figure]

The TBI parameters assessed along the same transect are in tight agreement with the previously reported results, particularly the findings by Kirwan et al. (2014), demonstrating the usefulness of the method to characterize the decomposition environment of tidal wetland soils.

L212-214: provide the data from Keuskamp et al. as well, we cannot compare or assess how much higher your data are.

Good idea, will do!

L 427-434: This is somewhat problematic also. It demonstrates the disadvantages of using these operationally defined indices; to which extent is this caused by the assumption that S is identical for the two types of substrate?

We agree that the operational definition can cause problems and its implications will be addressed as mentioned further up. The assumption that S is identical for the two types of substrate is irrelevant in this context: S is determined only based on what is left of the green tea material after incubation. The problem discussed in this section is that more material was decomposed from the green tea material than theoretically possibly based on its hydrolysable fraction. So either other, non-hydrolysable material was also decomposed to a considerable degree (as also mentioned by the reviewer further up), or the hydrolysable fraction is in fact higher than previously assessed.

Secondly, keep in mind that anaerobic decomposition processes are important in tidal wetlands, and can occur at high rates (similar order of magnitude as aerobic decomposition) up to substantial depths.

This is a valid point. We will briefly address this here and add citations illustrating that decomposition under anaerobic conditions can be equally high (but also equally low) as under aerobic conditions depending on the chemical quality of the substrate (Kristensen et al. 1995,

*Limnology and Oceanography*; Hulthe et al. 1998, *Geochimica et Cosmochimica Acta*; Kirwan et al. 2013, *Biogeosciences*; Mueller et al. 2016, *Global Change Biology*).

**INTERACTIVE COMMENT 1 by Dr J. Sarneel**
Have you thought about the fact that you are the first ones that tested the TBI method in saltmarsh systems? In fact, you are only the second to test it outside the pure terrestrial system (see Sarneel et al 2017 for riparian systems). Since many people are interested in experiences of using TBI outside the terrestrial system, this is valuable study, as it expands the range over which it was used. However, although understandable for this study, presenting k and S based on new hydrolysable fractions does not help direct comparison with the values obtained in other studies. Could you provide the data following the standard method of keuskamp et al 2013 in an appendix? I further wondered if you think that the salt water can have influenced what was the hydrolysable fraction?

We thank Dr Sarneel for the helpful comment on our study. We will add a line or two in the introduction reporting in which context the method has already found application. We will certainly also supply the data calculated in accordance with the original TBI protocol. Salinity of the systems did not occur to be an important predictor for S, so we think this wouldn't explain our negative S-values.

**References**

Blum, L. K. and Christian, R. R.: Belowground production and decomposition along a tidal gradient in a Virginia salt marsh, in The ecogeomorphology of tidal marshes, edited by S. Fagherazzi, M. Marani, and L. K. Blum, pp. 47–73, American Geophysical Union, Washington D.C., USA. [online] Available from: http://www.agu.org/books/ce/v059/CE059p0047/CE059p0047.shtml (Accessed 4 March 2014), 2004.

Craft, C.: Freshwater input structures soil properties, vertical accretion, and nutrient accumulation of Georgia and U.S tidal marshes, Limnol. Oceanogr., 52(3), 1220–1230, doi:10.4319/lo.2007.52.3.1220, 2007.

Djukic, I., Kepfer-Rojas, S., Schmidt, I. K., Larsen, K. S., Beier, C., Berg, B., Verheyen, K. et al.: Early stage litter decomposition across biomes, Sci. Total Environ., 628–629(January), 1369–1394, doi:10.1016/j.scitotenv.2018.01.012, 2018.

Hulthe, G., Hulth, S. and Hall, P. O. J.: Effect of oxygen on degradation rate of refractory and labile organic matter in continental margin sediments - A comparative survey of certain organic and inorganic compounds in an oxic and anoxic Baltic basin, Geochim. Cosmochim. Acta, 62(8), 1319–1328 [online] Available from: http://www.ingentaconnect.com/content/els/00167037/1998/00000062/00000008/art00044%5Cnhttp://dx.doi.org/10.1016/S0016-7037(98)00044-1, 1998.

Kirwan, M. L., Langley, J. A., Guntenspergen, G. R. and Megonigal, J. P.: The impact of sea-level rise on organic matter decay rates in Chesapeake Bay brackish tidal marshes, Biogeosciences, 10(3), 1869–1876, doi:10.5194/bg-10-1869-2013, 2013.

Kirwan, M. L., Guntenspergen, G. R. and Langley, J. A.: Temperature sensitivity of organic-matter decay in tidal marshes, Biogeosciences, 11, 4801–4808, doi:10.5194/bg-11-4801-2014, 2014.

Kristensen, E., Ahmed, S. I. and Devol, A. H.: Aerobic and anaerobic decomposition of organic matter in marine sediment: Which is fastest?, Limnol. Oceanogr., 40(8), 1430–1437, doi:10.4319/lo.1995.40.8.1430, 1995.

Lovelock, C. E., Feller, I. C., McKee, K. L. and Thompson, R.: Variation in mangrove forest structure and sediment characteristics in Bocas del Toro, Panama, Caribb. J. Sci., 41(3), 456–464, doi:ISSN 0008-6452, 2005.

McKee, K. L., Cahoon, D. R. and Feller, I. C.: Caribbean mangroves adjust to rising sea level through biotic controls on change in soil elevation, Glob. Ecol. Biogeogr., 16(5), 545–556, doi:10.1111/j.1466-8238.2007.00317.x, 2007.

Morrissey, E. M., Gillespie, J. L., Morina, J. C. and Franklin, R. B.: Salinity affects microbial activity and soil organic matter content in tidal wetlands., Glob. Chang. Biol., 20(4), 1351–62, doi:10.1111/gcb.12431, 2014.

Mozdzer, T. J., McGlathery, K. J., Mills, A. L. and Zieman, J. C.: Latitudinal variation in the availability and use of dissolved organic nitrogen in Atlantic coast salt marshes, Ecology, 95(12), 3293–3303, 2014.

Mueller, P., Jensen, K. and Megonigal, J. P.: Plants mediate soil organic matter decomposition in response to sea level rise, Glob. Chang. Biol., 22(1), 404–414, doi:10.1111/gcb.13082, 2016.

Prescott, C. E.: Litter decomposition: What controls it and how can we alter it to sequester more carbon in forest soils?, Biogeochemistry, 101(1), 133–149, doi:10.1007/s10533-010-9439-0, 2010.

Weston, N. B., Dixon, R. E. and Joye, S. B.: Ramifications of increased salinity in tidal freshwater sediments: Geochemistry and microbial pathways of organic matter mineralization, J. Geophys. Res., 111(G1), G01009, doi:10.1029/2005JG000071, 2006.

---

## Author Comment (AC2) · 26 Feb 2018

We thank all three reviewers as well as the author of the Interactive Comment for their constructive comments and suggestions. In the Supplement to this post we respond to each of these separately. Unless stated otherwise, we intend to incorporate these changes into a final version of the manuscript.

Please also note the supplement to this comment:
https://www.biogeosciences-discuss.net/bg-2017-533/bg-2017-533-AC2-supplement.pdf

---

## Author Response (AR3)

This pdf file contains point-by-point replies and marked-up ms versions of
- the first round of reviews (major revisions)
- the editor response after revisions (minor revisons)

**1. Point-by-point reply to the reviewer comments**   (first round of reviews, major revision)

**2. Marked-up manuscript version (text only)**   (first round of reviews, major revision)

**1. Point-by-point reply to the reviewer comments:**

We thank all three reviewers for their constructive comments and suggestions. In the pages below we respond to each of these separately.

In the following, reviewer comments are shown in in black, author responses in red font. Line numbers given below refer to the revised version of the ms (separate file, not the marked-up version below).

**REVIEWER 1, anonymous**

**General comments**

This article deals with an important aspect of carbon's fate in coastal wetlands in relation to global changes and their impacts on these ecosystems. Indeed, wetlands are receiving a growing attention in the climate change debate in relation to their high capacity to sequester blue carbon. Ecosystems considered in this "global" scale study are mainly tidal marshes but some mangroves sites were counted in the selected sites. Authors are assessing OM degradation and transformation, as a proxy of Carbon sequestration using the TBI approach. Thus, authors claim that they provided indirect evidences that rising Temperature and Sea Level and eutrophication will impact the capacity of tidal wetlands to sequester carbon. This work is worthwhile to publish although as authors cautioned, there are limits with the used method (obvious quality differences of Tea-bag OM with "real" plants) and also that they may have missed some influent factors that control OM degradation and sequestration. Introduction was well thought and the methodology was clear however, some choices were not judicious in the context of this study and may need to be revaluated (see specific comments). The adding of TIDE experimental site was a very interesting. The discussion is well organised but it needs to be shortened.

We thank the reviewer for his constructive feedback on our study. As requested, we streamlined the discussion where appropriate; however, several required additions have also been made, so that the overall length of the Discussion did not substantially change.

**Specific comments**

I am not a specialist of meta-analysis, therefore I will not comment on the validity or not of the numerical methods, but one thing is sure, analyses need always to rely on field knowledge even if results are "counterintuitive". The discussion is based on two characteristics (k, S) that are related to the quality and the fate of the litter-bags contents (here Tea-bags) which are strongly related to sedimentation dynamic and water velocity. In absence of a clear indication on how sediments (and OM) are behaving in each site, I am concerned about the amalgam in the same meta-analysis different systems in term of hydrological functioning: Salt Marches vs. Mangroves, High tide vs. low tide (in salt marches). For instance, estuarine mangroves receive loads of sediments from rivers whereas Europeans salt marches in open Bays get sediments mainly from the oceans. One way to tackle this concern is to process the same calculations/test s/figures without adding the mangrove sites to the pool of data. Same thing can be done by considering the main origin of sediments (not to confound with OM), without impacted TIDE sites, river presence or not, water velocity, human activities: … . These factors, of ecological importance, might be those missing to explain some global, or local, differences. If these data
cannot be compiled they should at least be discussed.

We agree with the reviewer that factors other than those assessed in this study might have
been influential and could have masked expected results (i.e missing temp effect on k). In
accordance with the reviewer's suggestion, we elaborated on this in several sections of the
discussion: e.g. 392-395; 399-407.

Response concerning sedimentary factors:
We agree with the reviewer that the different systems we compiled in a single meta-analysis are
characterized by potentially important differences in both sediment load and origin. We did not
explicitly assess sediment loads of our study sites. However, by distinguishing between
minerogenic and organogenic systems (i.e. sediment rich vs. sediment poor systems) in our
analyses, we are confident to have already captured the relative importance of sediment load
on our response variables. Please note that this categorical factor did not show up to be
important in our classification- & regression-tree data mining (CART). Furthermore, our two
most important findings (i.e. S decreases with temperature; S is lower in low vs. high elevated
zones) are consistent within both minerogenic and organogenic systems.

We indirectly also addressed sediment origin (riverine vs. marine) by including both estuarine
and coastal systems in our study. Specifically, we tested for effects of salinity class (fresh,
brackish, salt) on our response variables, with fresh systems far up in the estuary experiencing
the lowest marine influence and salt-water systems experiencing the highest marine influence.
If sediment origin (riverine vs. marine) had an important influence on our response variables,
this should have been reflected in our meta-analyses (i.e. Table 2). That being said, salinity of
floodwater and sediment origin can of course not easily be separated in an observational study.
Concerning the reviewer's remark on the sediment origin of our mangrove sites, it needs to be
noted that those were not estuarine (as assumed by the reviewer) but coastal systems in the
present study.

**REVIEWER 2, Dr J. Keuskamp**

**General comments**
This paper discusses the control that the soil matrix exerts on the decomposition of organic
matter in tidal wetlands. Their large carbon stocks and sensitivity to global change make this a
highly relevant topic for scientists and policy makers alike. The paper is well-written and easy to
read, while presenting novel data with important conclusions on the relation between
decomposition and global change. The usage of a standardised method over a wide range of
tidal systems allows for a generalisation to the global scale, making this paper relevant to the
broad readership of Biogeosciences. The explorative nature of the experiment also introduced
some unavoidable methodological weaknesses. Many of the environmental parameters which
are discussed in relation to decomposition are often strongly correlated with tidal regime (i.e.
soil temperature, salinity, nutrient status, microbial biomass, and redox status), or latitude (i.e.
nutrient limitation, vegetation type). In its current version, the manuscript does not always acknowledge the potentially spurious relation between these factors. While this does not
invalidate the main conclusions I would recommend to consider non-causality more carefully
when attributing effects to specific environmental parameters.
We thank Dr Keuskamp for his constructive comments on our study. We agree that correlations
between the assessed environmental parameters should be carefully considered in the
interpretation of our results. Accordingly, we put more emphasize on this throughout the ms;
some examples from different sections:
Methods/statistics:
"As we did not expect temperature to be independent of other parameters in this observational study, we constructed
a Spearman correlation matrix including the parameters temperature, latitude, tidal amplitude, salinity class, $k$, and $S$.
Additionally, we tested for differences in these parameters between marshes and mangroves and sites with mineral
and organic soils, using Mann-Whitney U tests (Table 2)."
Results:
"Temperature was highly correlated with latitude and tidal amplitude, and temperature was not independent of soil
type (mineral/organic) and ecosystem type (marsh/mangrove) (Table 2). The effect of latitude was similarly
pronounced as the temperature effect on $S$ – and consequently –effects of these two parameters on $S$ cannot be
separated (Table 2). By contrast, tidal amplitude and soil type did not significantly affect $S$, and the difference in $S$
between mangroves and marshes was only marginally significant (Table 2). These findings suggest that the presented
temperature effect on $S$ occurs to be mainly independent of tidal amplitude and soil type."
Discussion:
"Future experimental work is therefore required in order to further assess the effects of temperature on OM
stabilization and to separate temperature from latitudinal and other interacting effects (e.g. as outlined above for k)
that are difficult to control for in observational studies."
The reviewer is specifically concerned about interactions with the parameters tidal regime and
latitude.
→In terms of describing the tidal regime, we assessed tidal amplitude and, by comparing high
and low elevated zones within sites, a relative measure for flooding frequency (i.e. low zones
more frequently flooded than high zones). Tidal amplitude did not affect k and S (Table 3). It
showed up as a potentially important predictor in our CART, probably because of its strong
correlation with other parameters. However, this result needs to be considered cautiously
because splits based on tidal amplitude suggest mixed effects (Fig. S1a).
→In terms of flooding frequency (high vs. low elevated zones), we discuss a number of
potentially relevant interactions that were mentioned by the reviewer: redox -> 423-434;
salinity -> 444; nutrient status -> 452-465. The reviewer makes an important point by
mentioning soil temp interactions with tidal regime. We will address this point further below
where soil vs air temp differences are discussed.
→Changes in nutrient status/limitation and vegetation type with latitude are relevant for the
interpretation of the temperature effects on S and k. We accordingly extended discussion of
temperature effects on k and S: 392-395; 399-407
The current description of the data-analysis does not describe how the authors have
ascertained themselves that underlying assumptions of the statistical tests used were not violated. Where applicable, tests of heterogeneity, normality, and independence should be
included, or other tests considered.
The reviewer is correct. We revised the description of the statistics and also corrected some
statistical analyses:
Specified assumption checks:
"To test for effects of relative elevation (as proxy for relative sea level) on k and S, two-tailed paired t-tests were
conducted. Mean values of high and low elevated zones of the 21 sites where tea bags were deployed in both high
and low elevation zones were used (n = 21). The absence of outliers and normal distribution of the difference in the
independent variable (as assessed visually) assured robustness of paired t-tests. To assess the consistency of potential
effects of relative elevation on k and S, one-way ANOVAs were used in each site separately (replication was
sufficient in 20 sites). Normal distribution of residuals was assessed visually, Levene's test was used to test for
homogeneity of variance, and data were log-transformed if assumptions were not met. Mann-Whitney U tests were
conducted as a non-parametric alternative when log-transformed data did not meet ANOVA assumptions (Table S2).
We tested for effects of nutrient enrichment on k and S in the data from the TIDE project site (Massachusetts, US)
using two-way ANOVA with enrichment treatment and marsh zone as predictors. When Levene's test indicated
heterogeneous variance (true for k), data were log-transformed, which stabilized variance. Normal distribution of
residuals was assessed visually."
Corrected statistics/data structure:
Statistics in Table 2 (Spearman correlations and U tests) were based on mean values of each
site-by-zone combination (n = 51). Sites with observations in two zones were thus
overrepresented. In the revised version these statistics are based on site means (n = 30;
compare 2.2).
A related point is that the three sites at the Ebro delta (and the three Maine sites) were
considered as different zones of the same site, characterized by  different salinities
(fresh/brackish/salt marsh). However, we noticed that they are actually as far apart as the two
sites in Massachusetts or the three mangrove sites in Panama. For reasons of consistency, they
are now considered separate sites. This, however, does not change any of the
findings/conclusions previously drawn. We clarify in the Methods that many of our sites are co-
located in larger estuarine/coastal regions (line 148 and revised Table 1).
For example a linear fitting is performed between k and S with temperature, without
mentioning testing for residual patterns to uncover non-linearity. As the authors note the
relation between decomposition and single parameters are often not linear (L221), in which
case the result of a linear model is unreliable.
The reviewer is correct. A linear effect of temp is not expected. The intention for showing the
linear fit was only to better illustrate the significant temp effect on S (as tested/identified with
non-parametric Spearman correlation). However, we should not have used linear regression as
an additional hypothesis test. → We only use Spearman correlation for this in the revised
version. Yet, to better illustrate the temp effects, we still present scatterplots and use curve
fitting to illustrate significant temp effects. Indeed, the model with both highest $R^2$ and lowest
standard error of estimate describing the significant temp effect on S is not linear but
logarithmic (Figure 2)

Lastly, I would like to add that the strength if the TBI lays in its standardisation. I would
therefore recommend to mention the S/k calculated with the standard approach alongside with
the re-scaled values calculated with the more aggressive extraction method. This would allow
for easy comparison with other data such as the TBI-values from mangroves mentioned in the
methods paper. See also below.
We agree with the reviewer. The same point has been raised by Dr Sarneel in an interactive
comment (below). We have prepared a table with all site x zone values for k and S, giving both
the original TBI-values and the modified (Table S3; referenced in the ms: L513).
**Specific comments**
L79 and L83-L84 seem largely redundant to me
Deleted old lines 79-80
L85-L86 'OM decomposition' is somewhat ambitious as it is not clear whether this refers to
decomposition rate (k) or extend (S), please revise.
The sentence was changed to "Consequently, global changes that might decrease OM preservation in
tidal wetland soils not only affect carbon sequestration, but also decrease ecosystem stability against
SLR." Obviously, preservation is also affected by decomposition rate and stabilization; however,
we do not intend to specify the processes at this stage of the Introduction, but do this further
down in the text (i.e. 115-119).
L117 Although this should have been more explicit in the TBI method paper (Keuskamp et al,
2013), the k estimated by TBI is not exactly equivalent to the classical litter bag experiment as it
describes the decomposition rate of the hydrolysable fraction and is not calculated over the
entire mass. We have therefore adapted k1 to indicate that this is the k of the most labile
fraction, as opposed to k2 which refers to the decomposition rate of the recalcitrant fraction. To
avoid confusion this should be made explicit here.
We avoid reference to classical litter bag experiments here and instead make the meaning of *k*
clearer in the respective section of the Methods.
L120 The recalcitrant fraction is also decomposable, albeit a lot slower
This was poor wording of course →changed to "rapidly decomposable".
L127 ' thereby improving our process-level understanding on how global warming affects
carbon turnover' Not sure what this means exactly
Deleted "process-level".

L137 I am somewhat surprised that the oxidation of organic matter would be limited by the
supply of SO4 in brackish tidal wetlands. Wouldn't the constant flushing with water replenish
SO4 to saturating levels in brackish/salt water systems?

Well, it probably depends on how much seawater input the brackish system experiences.
Anyhow, our dataset does not actually allow to accurately describe salinity effects on k and S
(too imbalanced, low number of fresh systems). We simplified accordingly and only test for
effects of temperature, relative sea level, and eutrophication. Compare 120-135, 240-265

L154 '(i.e. dwarf vs. fringe phenotypes)' Aren't these also Rhizophora vs Avicennia? In that case
phenotypes would not be the appropriate description. These mangroves belong to different
genera, each with their own properties (soil oxygenation, phenolic compound production, N-
content) that are known to influence decomposition.

In most cases you would assume so, but here both fringe and dwarf are indeed Rhizophora with
very few individuals of Avicennia: Please compare: Mckee et al. (2007) Global Ecology and
Biogeography, **16**, 545–556; Lovelock et al. (2005) Caribbean Journal of Science, Vol. 41, No. 3,
456-464, 2005

L154 'Relative elevation' as relative to what? mean lower tide, mean mean tide? Please specify

We specified as follows:
"In 21 sites, we compared high and low elevated zones, which were characterized by distinct plant species
compositions (i.e. different communities in high vs. mid vs. low marshes) or by different stature of
mangroves (i.e. dwarf vs. fringe phenotypes). We used relative elevation (i.e. high vs. low elevated zone)
as a site-specific proxy for relative sea level. By doing so, we did not capture the actual variability in the
tidal inundation regime across our study sites as these vary in absolute elevation and in elevation relative
to mean high water."

L169-170 Decomposition rates depend on soil temperature rather than on air temperature.
Others have shown (e.g Piccolo et al. 1993, Reckless et al. 2011) that in tidal wetlands, the soil
temperature is strongly determined by inundation regime in which case the accuweather
temperature are not an accurate reflection of the decomposition environment. Moreover,
inundation regime and temperature effects would be confounded. Could it be shown
accuweather estimated temperatures vs measured temperatures so that the reader can see for
themselves whether the accuweather approximation suffices?

Dr Keuskamp brings a valid point here that indeed needs more consideration. Air temperature
would obviously diverge from soil temperature depending on factors such as canopy shading or
tidal regime and water temperature. As a consequence, air temperature can only approximate
the temperature conditions of the actual decomposition environment. However, considering
that we stretch a temp gradient of approx. 20°C, we are confident that this would also translate
into a profound soil-temperature gradient across our study sites.

The two studies mentioned by the reviewer, Piccolo et al. 1993 and Ricklefs et al. 2012, present
data for un-vegetated tidal flat systems. For marsh systems, we would rather refer to Kirwan et al. (2014). The authors show, that in marshes along the well-studied latitudinal gradient of the
US East coast (and we do share a number of sites), soil temp and air temp are highly correlated,
while the relationship between soil temp and water temp is far weaker (Kirwan et al. 2014,
*Temperature sensitivity of organic-matter decay in tidal marshes*; biogeosciences: **Fig. 2a**).
In our study sites, we did not continuously measure soil temp over the 3 months of deployment,
and thus it is difficult to assess how well soil and air temp were correlated in this study.
However, in several of our sites, soil temp was assessed at the time point of insertion and
retrieval of bags. We plotted these data against the mean air temp of the day as acquired from
the *accuweather* service in Figure S3. → We see that generally air temp is a good proxy for soil
temp across sites. Yet, there is considerable variability in soil temp not explained by air temp,
which would result from the fact that soil temp was assessed in one time point as opposed to
mean air temp of a single day and of course from other factors, such as distance of weather
station from site, shading, influence of water temp etc..
→We agree with the reviewer that this needs to be stated and discussed in the manuscript.
Accordingly, we put more emphasize on this throughout the ms; some examples from different
sections:
Methods:
"It needs to be noted here, that top-soil temperature would differ from air temperature depending on factors such as
canopy shading or tidal regime and water temperature. As a consequence, air temperature can only approximate the
temperature conditions of the actual decomposition environment (Fig. S3)."
Discussion:
"The present study used air temperature as a proxy for top-soil temperature. Thus, the temperature regime of the
decomposition environment was only approximated, which certainly would have weakened a significant relationship
between temperature and k. However, following typical Q10 values for biological systems of 2-3 (Davidson and
Janssens, 2006), k should have at least doubled over the gradient of $\Delta T >15°C$; yet our data do not even show a
tendency of an effect ($rs = 0.02$; Table 2). We therefore propose that other parameters exerted overriding influence
on k, mainly masking temperature effects, and have not been captured by our experimental design. This notion is in
line with the fact that studies conducted at …"
Lastly, we want to stress a related point here: "low" and "high" in the figure legend of Fig.S3
refer to the low and high elevated zones within the systems. A paired t-test comparing the
difference of air temp and soil temp between the paired high and low elevated zones indicates
no significant effect of zone ($p = 0.563$). This shows that differences between air and soil temp
were not consistently more pronounced in either the low or the high elevated zones.
Additionally, soil temp was not significantly affected by zone ($p = 0.342$). One of our main
findings, that S is consistently lower in low vs. high zones, is consequently not temperature
affected (i.e. *S* was significantly reduced in 14 of 20 sites, and the opposite was observed in
none of the sites (Table S2).
L176 Pepsico, to my knowledge the bags are produced by Lipton, which is a Unilever brand.
Unilever belongs to PepsiCo, but of course the tea is produced by Unilever. PepsiCO was deleted
in order to avoid confusion.

L180 Were the reference bags dried at 70oC prior to mass determination?

This may be a misunderstanding: reference bags were used to determine a mean value of the empty nylon bag itself without contents. I do not know if that one has always been dried, however, empty-bag weights were very consistent among labs. →By contrast, initial tea-content weights showed some variability across the involved labs. I also noticed that some labs, after drying at 70°C, used desiccators, in which the material could cool down without sucking moisture, before weighing and some didn't. I therefore assessed if potential moisture differences of the initial tea material or differences in the amount of the initial material could have affected S or k. However, there was no relationship between green initial weight and S ($r2=0.0003$; $p=0.936$) and no between rooibos initial and k ($r2=0.005$; $p=0.728$).

L198-L200 It could well be that the method described is a more accurate operationalisation of the labile (non-hydrolysable) fraction. Redefining the labile fraction and the consequential shift in S, and rescaling of k, may however lead to misunderstandings when the results of this study are used in comparisons with other TBI experiments. I would therefore suggest to provide the TBI S/k values calculated according to protocol alongside the obtained S/k values obtained by the revised protocol.

We agree. We added a table with the original TBI values accordingly (see comment further up).

L220-L250 Would you be able to indicate whether potential violations of the assumptions underlying the statistical tests were assessed? For example, were the residuals of the ANOVA procedure tested for normality / homogeneity of variance?

This was indeed missing. We added these missing details to the Methods, see comment further up.

L250 It is critical to this conclusion that air temperature is a good proxy of soil temperature (see earlier remark). The interaction between temperature effect and tidal position reinforces the suspicion that this is not the case.

We agree with the first half of this remark (see addressed further up), but not with the second. That is, there is no clear interaction between tidal position and temperature: Temperature seems to affect k in mesotidal systems (tidal amp >2.1m) with k higher in sites with temp >14.5°C; however, this apparent temp effect is inconsistent within this group of mesotidal systems. That is, sites with temp >18.2°C show lower k than those sites with temp <18.1°C. Please note comment further up: temp did not differ between high and low elevated zone, neither did the temp difference between soil and air.

L314 As also noted in L313, the absence of a temperature effect is very unusual. Could the authors rule out the possibility that this is due to a mismatch between soil and air temperature?

We stretch large gradients of approx. 20°C for both soil and air temp, and there is not even the
slightest tendency of a temp effect on k (Spearman's rank coefficient = 0.02; Figure 2), while S is
strongly affected. It therefore seems that that other factors exert overriding control over k and
more strongly mask temp effects on k than on S. Yet, we agree on the need to discuss the
methodological inaccuracy in determining temp of the decomposition environment, and we
addressed this point (see comment further up).
We want to stress a related point here concerning the missing temp effect on k: In order to
address remarks by Reviewer 3 and demonstrate the usefulness of the TBI method for tidal
wetlands, we took a separate look at the data of the North American East coast latitudinal
gradient along which previous studies have shown clear temp effects on decomposition
processes and microbial biomass (Blum et al. 2004; Kirwan et al. 2014; Mozdzer et al. 2014).
Species composition of these marshes is quite constrained (i.e. Spartina alterniflora dominated)
reducing confounding effects induced by differences in vegetation. Along this gradient, we
clearly see an increase in S and also the expected decrease in k, although temp explains more
variability for S (Fig. S2). We added this figure to the manuscript in order to illustrate that temp
effects on k can be identified on the regional scale, but not on the global scale with more
confounding factors.
The effect of temp on k at the regional scale but the missing effect at global scale is also in
agreement with the just recently published article on *Early stage litter decomposition across*
*biomes* by Ika Djukic and others. (Although they did not assess specifically k and S in their study
using the TBI tea materials, they simply assessed mass loss of the two materials). Across biomes,
climate (temp and precipitation) had no effect on break down; however, within biomes the
effect was strong.
L332 I would recommend discussing potential confounding of temperature effects with other
changes in decomposition matrix (e.g. nutrient availability, redox status, vegetation, salinity).
With respect to k, such reservations are made in L323/L329, but are absent here.
We agree with the reviewer and added similar considerations for the discussion on temp effects
on S (400-407).
L351 Can this be generalised to continuously submerged parts of the soil? The TBI is at a
relatively low depth, where tidal pumping may cause increased influx of oxygen during tidal
subsidence. Especially in tannin-rich mangrove systems, temporal oxygenation may make a
large difference by allowing breakdown of phenolic compounds (see also Freeman et al, 2001)
We agree with the reviewer. We elaborated our discussion on expected redox effects, also with
respect to comments by Reviewer 3 (427-435, 515-521).
L445 In mangrove TBI experiments that I have conducted S values have always been positive,
and I am somewhat puzzled by the large difference. Negative S values could also be caused by
loss of recalcitrant particles as I have observed when using teabags in open water. Did you have
any indications that this has taken place here?

We were puzzled as well when realizing that so many values were lower than they should be.
Indeed, the FL mangrove values you report in Keuskamp et al 2013 are considerably higher.
That's also when I decided to check whether the quality of the material had changed.
→No, I am not aware of loss of particles from the bags in situ. In fact, in a recent study
(microcosm study, Wadden Sea) we used the new tea bags (those without nylon mesh) that
wouldn't allow for loss of material through the mesh. Also with these bags, we had ~11%
negative values (Hao Tang, Peter Mueller et al. unpublished data), comparable to what we
found in some Wadden Sea marshes in the present study using nylon mesh bags.
→Comparing our results to those reported in Djukic et al. (2018), it becomes clear that negative
S values occur less frequently across terrestrial systems, however, are not negligible either.

Technical corrections L74 Earth? Not sure if this should be with a capital E
L77 Separate SRL from citations
L94-98 This sentence is very hard to read. Split.
L346 add 'in' before 'tidal wetlands'
Thanks, technical corrections have been made.

**REVIEWER 3, anonymous**

Mueller et al. conducted decomposition experiments using tea bags based on a standardized
approach developed by Keuskamp et al. (2013), across different marsh and mangrove sites in
order to cover a gradient in temperature, inundation regime, etc. While such cross-ecosystem
studies have a high potential, I feel the impact of this dataset in terms of new insights is
relatively limited. The dataset can be published but I feel the impact of the conclusions should
be toned down somewhat – the manuscript does not really deliver what the title suggests. The
dataset should be publishable, but it needs a more critical discussion and should provide the
readers with a more complete overview of the caveats and assumptions used in the TBI
approach, so that the readers can better assess what can and cannot be deduced from these
data. My main point is that the TBI index – both the original and the modified protocol
suggested here – has plenty of limitations and it remains an operationally defined procedure,
with several assumptions that are open to discussion. In addition, we are not looking at
mineralization of in situ produced material hence some interactive effects will be missed in this
approach; results should not be over-interpreted or generalized.

We thank the reviewer for his critical and constructive feedback on our work. We have revised
our ms, particularly the discussion part (4.4 Methodological considerations; 4.5 Implications), in
order to provide the reader with a more complete overview of the assumptions and limitations
involved with the TBI approach. We have provided more detailed responses below regarding
the specific comments raised by the reviewer.

**Specific suggestions**

L55: "stabilization was 29% lower": this does not mean much if you do not define stabilization
here, it can be interpreted in different ways. For me this remains a somewhat problematic proxy
(see further comments).
We agree with the reviewer's concern and specified the parameter in the abstract (53-55). The
second part of the comment will be addressed further below.
L60-61: data from the eutrophication experiment: would not extrapolate this to 'high sensitivity
to global change'. Eutrophication will also affect the nutrient content of locally produced
biomass, this aspect is not taken into account when standardized material is used in the
experiments.
The reviewer is of course correct to state that with eutrophication, also the quality of the
biomass produced in the system would change with potentially important consequences for the
decay process. Thus, interpretation of the results obtained with standardized litter need to be
conducted cautiously. In the discussion it now reads as follows:
"Standardized approaches like this, or also the cotton-strip assay (e.g. Latter and Walton, 1988), are useful
to separate the effects of environmental factors other than OM quality on decomposition processes and to
assess their relative importance. Otherwise, complex interaction effects of the abiotic environment and
OM quality make it difficult to predict the relevance of certain environmental factors for decomposition
processes, potentially masking the effects of important global-change drivers (Prescott, 2010). At the same
time, however, the global-change factors considered in the present study are likely to induce changes in
the quality of the OM accumulating in tidal wetlands, for instance through shifts in plant-species
composition and plant-tissue quality, that can potentially counterbalance or amplify the effects on
decomposition processes suggested here. Future research therefore needs to address OM quality feedbacks
on decomposition processes in tidal wetlands in order to gain a more complete understanding of global-
change effects on tidal-wetland stability and carbon-sequestration capacity."
We agree with the reviewer that this sentence needs to be toned down in the abstract, because
there is no space for further elaboration on the assumptions and methodological
considerations. We will go with "potentially high sensitivity of OM stabilization to global change."
L90-95: an important caveat here is that you only study the decomposition of one type of
source material (well, in two versions), but not other sources that contribute to the OM pool
e.g. marine or other aquatic inputs into the intertidal system.
We agree; this is important for the interpretation of our results. However, conventional litter
bag experiments are also restricted in their choice of material; here actually lies an advantage of
the standardized approach, although we acknowledge that the quality of the TBI materials is
obviously closer to that of wetland plant litter than to the marine derived, labile allochthonous
organic input a tidal wetland receives. We have elaborated on this in 4.4:
"Interpretation of results obtained from standardized approaches like the present needs to be made
cautiously because OM quality (i.e. its chemical composition) is a key parameter affecting its
decomposition. As the quality of the TBI materials differ from that of wetland plant litters, and likely even
more from the quality of the imported allochthonous OM (Khan et al., 2011), we did not expect to capture actual rates of early-stage OM break-down in this study. Instead, we used the TBI to characterize the decomposition environment by obtaining a measure for the potential to decompose and stabilize the deployed standardized material."

Additionally, we toned down our Implications (4.5 ):

"This study addresses the influence of temperature, relative sea level, and coastal eutrophication on the initial transformation of biomass to SOM, and it does not encompass their effects on the existing SOM pool. However, aspects of S and k are key components of many tidal wetland resiliency models (Schile et al., 2014; Swanson et al., 2014) that have highlighted the critical role of the organic contribution to marsh elevation gain. Although actual rates of S and k cannot be inferred from this study using a standardized approach, our data identify strong negative effects of temperature, relative sea level, and coastal eutrophication on the stabilization of fresh organic inputs to tidal-wetland soils. We argue that these unanticipated combined effects yield the potential to strongly accelerate carbon turnover in tidal wetlands, thus increasing their vulnerability to accelerated SLR, and we highlight the need for experimental studies assessing the extent to which the here identified effects translate into native OM dynamics."

section 2.2: it is important for the readers not familiar with the Keuskamp et al. paper to re-iterate and stress the assumptions on which this approach is based, e.g. that k2 (decomposition constant of the non-labile fraction) is assumed to be 0, and that S is assumed to be similar for both types of tea. I still find this major shortcomings- we know the first assumption not to be valid, and I have not seen strong arguments to support the second assumption. The main reason to make these assumptions is to allow to estimate k and S using only one time point of measurements instead of having to measure at different points in time. These aspects deserve to be mentioned explicitly and the limitations of the approach should be discussed more critically. -What is the added value of this approach compared to simply measuring the decay of the biomass over a limited number of time steps, and using a more realistic decay model?

Please note that Keuskamp et al. (2013) show that the TBI is quite robust against deviations from the assumption that S is the same for the two materials. The assumption that k2 could be considerably higher than 0 during 3 months of deployment has already been questioned by us (520-522). However, the reviewer brings important points here. In accordance, we elaborated on the description of the TBI calculations in the respective section of the Methods (2.2):

"Decomposition rate ($k$) and stabilization ($S$) were assessed following the TBI protocol (Keuskamp et al., 2013). The TBI approach can be considered as a simplified litter-bag approach, allowing a time- and cost-efficient characterization of the decomposition environment, because $k$ and $S$ can be estimated without repeated sampling of the decomposing material as in conventional approaches. This implies the assumptions that (1) $S$ is equal for the two types of material used in the approach and (2) that decomposition of non-hydrolyzable materials during the 3 months of deployment is negligible. We refer the reader to Keuskamp et al. (2013) for further detail and validity assessments of assumptions."

The advantage of the TBI approach over a longer-term litter experiment is the time efficiency that allowed us to assess decomposition in a large number of sites during the same growth season and find enough collaborators capable to contribute with their work. Obviously, as outlined by the inventers of the method (Keuskamp et al. 2013), the TBI can't substitute the precision of classic litter bag methods, but it considerably reduces the effort necessary to fingerprint local decomposition. It is a trait-off between precision and effort that helps
gathering decomposition data across ecosystems and biomes.
In order to demonstrate the usefulness of the method and its comparability to other methods
assessing decomposition processes tidal wetlands, we will separately present our data on k and
S for the sites along the North American East coast latitudinal gradient, along which previous
studies have shown clear temperature and latitudinal effects on decomposition processes. For
instance, Kirwan et al. (2014; Biogeosciences) demonstrated a strong increase in cellulose decay
with both temperature and latitude, and Mozdzer et al. (2014; Ecology) showed a marked
decrease in sulfate reduction with latitude along this transect. The TBI parameters assessed
along the same transect are in tight agreement with the previously reported results, particularly
the findings by Kirwan et al. (2014), demonstrating the usefulness of the method to characterize
the decomposition environment of tidal wetland soils.

[Figure]

Figure S2 Site means of decomposition rate (left) and stabilization (right) versus mean air temperature of the deployment period shown for the ten sites situated along the latitudinal gradient of the North American Atlantic coast; state acronyms are shown (compare Table 1). Regression lines illustrate significant relationships^; regression models with lowest standard error of estimate (SEE) and highest $R^2$ are shown. Decomposition rate: $y = 0.001x - 0.0091$; $R^2 = 0.692$; SEE = 0.003; stabilization: $y = -0.712\ln(x) + 2.2331$; $R^2 = 0.860$; SEE = 0.070

Lastly, we discuss the applicability of the TBI approach in 4.4:
"Future research will have to test the applicability of the TBI approach in different ecosystems and test the
validity of its assumptions (i.e. S is equal for both types of material used, and mass loss of non-
hydrolyzable material is negligible over 3 months of deployment). The results of our regional scale
assessment along the North American Atlantic coast transect are in tight agreement with previously
reported results on cellulose break-down and soil microbial activity along this well studied transect
(Kirwan et al., 2014; Mozdzer et al., 2014). We can thereby demonstrate the usefulness of the TBI
approach to assess early-stage decomposition in tidal-wetland soils."
L212-214: provide the data from Keuskamp et al. as well, we cannot compare or assess how
much higher your data are.

Good idea, both are included now (Table S1).

L 427-434: This is somewhat problematic also. It demonstrates the disadvantages of using these
operationally defined indices; to which extent is this caused by the assumption that S is identical
for the two types of substrate?

We agree that the operational definition can cause problems and its implications have been
addressed as mentioned further up. The assumption that S is identical for the two types of
substrate is irrelevant in this context: S is determined only based on what is left of the green tea
material after incubation. The problem discussed in this section is that more material was
decomposed from the green tea material than theoretically possibly based on its hydrolysable
fraction. So either non-hydrolysable material was also decomposed to a considerable degree (as
also mentioned by the reviewer further up) or the hydrolysable fraction is in fact higher than
previously described.

Secondly, keep in mind that anaerobic decomposition processes are important in tidal wetlands,
and can occur at high rates (similar order of magnitude as aerobic decomposition) up to
substantial depths.

This is a valid point that needed more consideration. We specified in section 4.2:

[revised manuscript text omitted]

**Contents:**

**1.)** **Point-by-point reply**    (to editor response after revision)

**2.)** **Marked-up version**    (to editor response after revision)

**1.)    Point-by-point reply**

**EDITOR COMMENT:**    Dear authors, after reading your revised MS, I find it can be published in Biogeosciences after you consider the following technical point: In its present form, your MS is very short and the supplementary material relatively long. The MS that is supposed to be published contains only 3 figures. However, some figures in the supplementary material are cited a lot in the text: Fig S1 is cited 9 times and Fig S2 is cited 5 times. I wonder what are the motivations for this choice and if the paper would not benefit from the insertion some of this additional figures in the MS rather than in the supplementary material.

Could you please explain the motivation for this choice, unusual for Biogeosciences, and if relevant, provide a revised MS that include the most cited figures and tables in the main text.

**REPLY:**  Dear Editor, we considered figures 1 and 3 as well as table 1 to be quite large and therefore decided to move less relevant information to the supplement. In fact, we thought our MS would be rather too long than too short; so thanks for the rectification! We agree that some of the supplementary material is indeed cited quite often throughout the MS.

Concerning Fig. S1: This figure presents (only) the results of a data mining approach, and it is used only in addition/to support the results presented in the other figures and tables. We don't want the reader to think that it displays the primary output of our statistical analyses to test for temperature/sea level/eutrophication effects. Although insightful, it is less relevant for our story line than the other figurers/tables. That being said, we think it has been cited more often than necessary in the previous version of the MS. We addressed this, reducing the number of citations from 9 to 4.

Instead of moving Fig S1 to the main text, we would prefer to include Fig S2. Its results are indeed quite central, and inclusion would help the reader follow our story more easily. In the revised version, it is included as the new Fig 3 (of 4).

2.) MARKED-UP VERSION:

[revised manuscript text omitted]